# hReg-CNCC reconstructs a regulatory network in human cranial neural crest cells and annotates variants in a developmental context

Zhanying Feng [1,2], Zhana Duren[3,4], Ziyi Xiong[5,6,7], Sijia Wang [8,9], Fan Liu [7,10 ✉], Wing Hung Wong [4 ✉] & Yong Wang [1,2,9,11 ✉]

Cranial Neural Crest Cells (CNCC) originate at the cephalic region from forebrain, midbrain and hindbrain, migrate into the developing craniofacial region, and subsequently differentiate into multiple cell types. The entire specification, delamination, migration, and differentiation process is highly regulated and abnormalities during this craniofacial development cause birth defects. To better understand the molecular networks underlying CNCC, we integrate paired gene expression & chromatin accessibility data and reconstruct the genome-wide human Regulatory network of CNCC (hReg-CNCC). Consensus optimization predicts high-quality regulations and reveals the architecture of upstream, core, and downstream transcription factors that are associated with functions of neural plate border, specification, and migration. hReg-CNCC allows us to annotate genetic variants of human facial GWAS and disease traits with associated cis-regulatory modules, transcription factors, and target genes. For example, we reveal the distal and combinatorial regulation of multiple SNPs to core TF *ALX1* and associations to facial distances and cranial rare disease. In addition, hReg-CNCC connects the DNA sequence differences in evolution, such as ultra-conserved elements and human accelerated regions, with gene expression and phenotype. hReg-CNCC provides a valuable resource to interpret genetic variants as early as gastrulation during embryonic development. The network resources are available at https://github.com/AMSSwanglab/hReg-CNCC.

[1] CEMS, NCMIS, MDIS, Academy of Mathematics and Systems Science, National Center for Mathematics and Interdisciplinary Sciences, Chinese Academy of Sciences, Beijing, China. [2] School of Mathematics, University of Chinese Academy of Sciences, Chinese Academy of Sciences, Beijing, China. [3] Center for Human Genetics, Department of Genetics and Biochemistry, Clemson University, Greenwood, SC, USA. [4] Department of Statistics, Department of Biomedical Data Science, Bio-X Program, Stanford University, Stanford, CA, USA. [5] Department of Genetic Identification, Erasmus MC University Medical Center Rotterdam, Rotterdam, Netherlands. [6] Department of Epidemiology, Erasmus MC University Medical Center Rotterdam, Rotterdam, Netherlands. [7] CAS Key Laboratory of Genomic and Precision Medicine, Beijing Institute of Genomics, Chinese Academy of Sciences, Beijing, China. [8] Key Laboratory of Computational Biology, CAS-MPG Partner Institute for Computational Biology, Shanghai Institutes for Biological Sciences, Chinese Academy of Sciences, Shanghai, China. [9] Center for Excellence in Animal Evolution and Genetics, Chinese Academy of Sciences, Kunming, China. [10] China National Center for Bioinformation, Chinese Academy of Sciences, Beijing, China. [11] Key Laboratory of Systems Biology, Hangzhou Institute for Advanced Study, University of Chinese Academy of Sciences, Chinese Academy of Sciences, Hangzhou, China. ✉email: liufan@big.ac.cn; whwong@stanford.edu; ywang@amss.ac.cn

In the past decades, genome-wide association studies (GWAS) have identified many genetic changes for human facial variations and craniofacial defects[1–7]. These genetic variants are enriched in enhancers preferentially active in cranial neural crest cells (CNCC) and embryonic craniofacial tissue[2,4]. CNCC is a migratory cell population in early human craniofacial development that gives rise to the peripheral nervous system and many non-neural tissues such as smooth muscle cells, pigment cells of the skin, and craniofacial bones. Thus, a synthesis of the population genetics and developmental biology holds great promise to understand how differences in DNA sequence alter gene regulation in a specific cellular context and determine differences in their phenotypes. In other words, understanding the gene regulation mechanism of neural crest induction, specification, and migration will help interpret genetic variants of facial phenotype and facial disease mechanism.

Considerable efforts have been made in non-human species to understand the gene regulation of neural crest and found many important regulators, regulations, and pathways. FGFs and WNT inhibitors, Tfap2, Sox2/3 were responsible for induction and formation of the neural plate border[8–13]. Foxd3, Ets1, and Snai1/2 were the neural crest specifier genes to establish neural crest identity[14]. Neural crest underwent migration through EMT program[15,16], driven by WNT signal and Snai1/2. At regulatory network level[17], knockdown technology was utilized to experimentally test the function of 50 genes to form a neural crest gene regulatory network of lamprey[18], which was refined by the time-series transcriptome analysis[19]. Moreover, Simoes-Costa et al.[20] used neural-specific enhancer to isolate pure neural crest subpopulation and knocked down the neural plate border genes and early neural crest specifier genes to construct a gene regulatory network of neural crest.

However, the existing regulatory network studies were performed in non-human species and far from complete. Most importantly, they ignored the important role of cis-regulatory elements (REs), in which genetic variants located and altered regulation. For a better understanding of how genetic variants affect human craniofacial traits and diseases, a genome-wide regulatory network with non-coding REs for the human neural crest is in pressing need. Timely, Prescott, et al. performed epigenomic profiling for the iPSC induced human CNCC[21] to study facial enhancers. Wilderman et al.[22] profiled multiple histone markers of chromatin activity as comprehensive functional genomics data and predicted chromatin states for 4.5–8 post-conception weeks of early human craniofacial development. These multi-omics data make inferring a comprehensive human regulatory network of cranial crest cell computationally a possible task.

Recently, we demonstrated that paired expression and chromatin accessibility (PECA) data modeling can provide a detailed view of how trans- and cis-REs work together to affect gene expression in a context-specific manner[23]. PECA was successfully applied to identify the master regulator in stem cell differentiation[24] and interpret RE for non-model organism[25]. PECA2 further extended PECA by removing the requirement of paired data from a diverse panel of cell types, so that inference of context-specific regulatory network was possible from paired data on just one sample. PECA2 has been used to reveal causal regulations for time course data[26]. Here, we started from PECA2 to integrate paired RNA-seq and ATAC-seq data and constructed a human Regulatory network of Cranial Neural Crest Cell (hReg-CNCC) by a consensus optimization model, which was proposed to integrate the multiple replicate data, leverage the biological and technical replicates, and obtain a high-quality network. This high-quality hReg-CNCC outperformed single, union, and intersection networks, was validated by the known CNCC pathways, and revealed regulatory architecture for CNCC development. hReg-CNCC allowed us to better interpret findings on human facial variation such as various facial traits in GWAS Catalog, and cranial rare diseases. Further annotating evolutionary sequences with associated REs, transcription factors (TFs), and target genes (TGs) of hReg-CNCC also revealed important biological insights. These results demonstrated the capacity of hReg-CNCC in revealing the mechanisms underlying the genetic association observed for human craniofacial traits.

## Results

**Consensus optimization infers higher quality hReg-CNCC.** High-quality hReg-CNCC was constructed based on a two-step framework (Fig. 1). In the first step, we collected paired RNA-seq and ATAC-seq data from (Prescott, et al.)[21] and applied PECA2[26] to R replicates (R = 6, samples were matched at the cell type level, Supplementary Data 1) to obtain R context-specific regulatory networks (Methods). We defined cis-regulatory modules (CRM) associated with a TF-TG pair as a set of REs bound by TF to regulate TG. Then each network was pooled by TF-CRM-TG triplets with TF-TG regulatory strength ($S^r$, $r = 1, 2, \ldots, R$) and CRM ($C^r$, $r = 1, 2, \ldots, R$). In the second step, we developed a consensus optimization model to integrate the R regulatory networks ($S^r$ and $C^r$) and obtained hReg-CNCC with reliable regulatory strength $S$ and reproducible CRMs $C$ (Methods).

We validated hReg-CNCC using independent datasets from three different perspectives. Firstly, we collected the known CNCC pathways[17], including three CNCC functional modules and 95 regulations among 50 genes from non-human organisms, to assess our predicted TF-TG regulations. Genome-wide hReg-CNCC predicted 838,220 regulations among 450 TFs, 15,686 REs, and 7270 TGs. hReg-CNCC predicted 703 regulations among the 50 genes of CNCC pathway and 36 were in the known CNCC pathways. This gave recall rate 38%, precision rate 5.1%, and 0.09 F1 score. To test the robustness over parameter choices, we compared 6 hReg-CNCC versions (with different parameters in consensus optimization) with 6 single replicate inferred networks. hReg-CNCC obtained significantly higher values for precision (t-test P-value $4.6 \times 10^{-3}$), recall (t-test P-value $6.3 \times 10^{-3}$), and F1 score (t-test P-value $2.1 \times 10^{-3}$) (Fig. 2a). This demonstrated that our consensus optimization indeed handled the heterogeneity among replicates, integrated the varieties, and was robust to parameter choices. We used the highest F1 score to select the final consensus regulatory network hReg-CNCC from different parameters (Supplementary Fig. 1a, b, c), which was used in the following analysis. Next, we compared our consensus optimization with two naïve integration methods, i.e., union and intersection (Methods). hReg-CNCC showed better precision than union and intersection networks by 0.6% and 1.2%. For recall rate, hReg-CNCC achieved a delicate tradeoff between union network (about 50% recall with the largest coverage) and intersection network (about 10% recall), by greatly improving coverage while avoiding many false positives. As expected, hReg-CNCC achieved the best F1 score (Fig. 2b). Furthermore, we separated the three CNCC functional modules in CNCC pathways respectively and found hReg-CNCC covered more than half of known regulations. The coverage came to nearly 60% for "neural crest specification" and "neural crest migration" (Supplementary Fig. 1d). Then we collected another gene regulatory network as the gold standard for parallel validation, which was built with multi-omics data in chick[27]. We reached the same conclusion as with CNCC pathway: hReg-CNCC was significantly better than single networks for precision, recall, and F1 score (Supplementary Fig. 1e). Compared with the overlapping and union method, hReg-CNCC obtained the best precision, with a

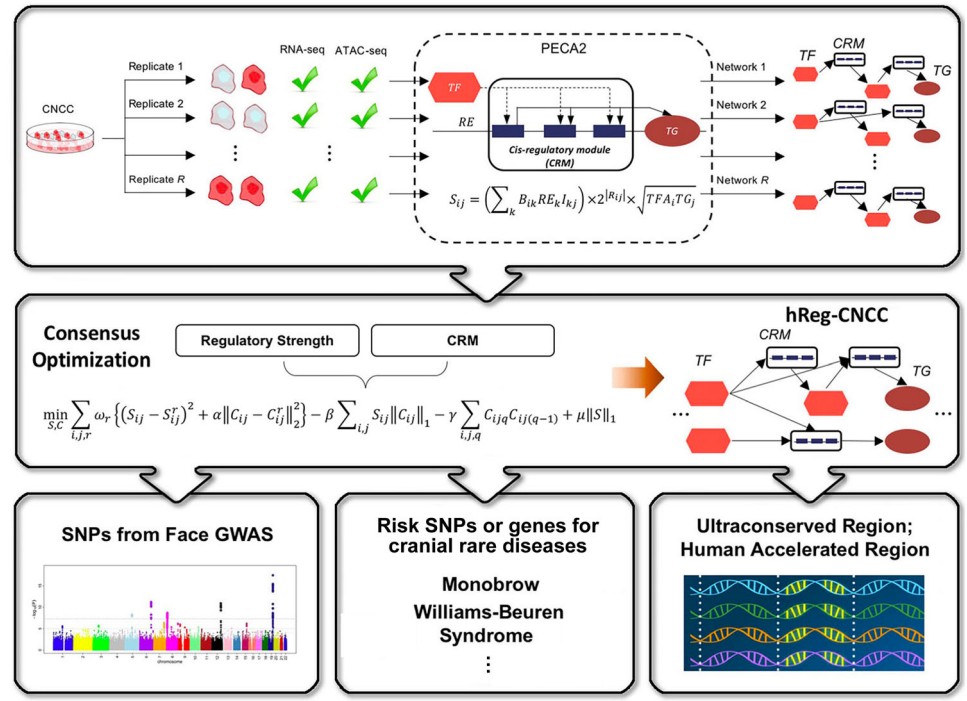

**Fig. 1 Schematic overview of inferring human regulatory network of CNCC (hReg-CNCC) based on paired gene expression and chromatin accessibility data.** High-quality hReg-CNCC is reconstructed by a two-step framework. Step 1: PECA2 infers context-specific regulatory networks from biological replicates (see "Methods" for details). Specifically, R CNCC replicates, each with paired RNA-seq and ATAC-seq data, are input into PECA2 and output R regulatory networks. We defined the CRM associated with TF - TG pair as a set of REs bound by TF to regulate TG. Each network is denoted by TF-TG regulatory strength ($S^r$, $r = 1, 2, \ldots, R$) and CRM ($C^r$, $r = 1, 2, \ldots, R$), i.e., the TF-CRM-TG triplets. Step 2: Consensus optimization integrates R regulatory networks ($S^r$ and $C^r$) and outputs hReg-CNCC with reliable regulatory strength S and reproducible CRMs C. hReg-CNCC serves as a valuable resource to interpret genetic variants from face GWAS, comparative genomics, and disease studies.

trade-off of recall. And hReg-CNCC performed best for F1 score (Supplementary Fig. 1f), which again showed hReg-CNCC was the best among the three methods.

Secondly, we used CNCC's context-specific ChIP-seq binding data[21] for master regulators to validate our TF-CRM regulations in hReg-CNCC. We processed the ChIP-seq data of *TFAP2A* and *NR2F1* and treated them as the gold standard to examine their predicted binding sites. In total there was 11,515 REs predicted to be bound by *TFAP2A* in hReg-CNCC and 80% of those REs overlapped with *TFAP2A*'s ChIP-seq peaks. This outperformed single networks with average precision of 0.76 and again demonstrated the advantage of integrating replicates (Fig. 2c). For *NR2F1*, there were 10,636 binding REs in hReg-CNCC and the precision was 0.57, which also outperformed the average precision for single networks 0.53 (Fig. 2c). Taken together, hReg-CNCC provided accurate TF binding predictions.

Thirdly, we tested the accuracy of our CRM-TG regulations in hReg-CNCC by TG's expression. We utilized the linkages between human biased enhancers[21] and human biasedly expressed genes as gold standard positives. For these human biased enhancers, hReg-CNCC predicted 216 genes as their TGs, of which 45 genes were human biasedly expressed genes. This gave a fold change enrichment 2.31 (Methods). We compared with Activity-By-Contact (ABC) model[28] and proximity-based method, which assigns the nearest TSS as TG (Methods). For ABC model, there were 260 genes that were predicted to be regulated by human biased enhancers and 38 of them were human biasedly expressed genes, which gave the fold change 1.60. For the proximity-based method, there were 1445 nearest genes linked to these human biased enhancers and 214 genes of them

were human biasedly expressed genes, which gave the fold change 1.62 (Fig. 2d). These results showed hReg-CNCC was more accurate to assign correct TGs for REs. Importantly, 15 hReg-CNCC predicted human biasedly expressed genes (33%) were regulated by distal enhancers and cannot be correctly predicted by ABC model or proximity-based method (Fig. 2e). For example, *ROBO3*, which confined early neural crest cells to the ventral migratory pathway in the trunk[29] and regulateed the production of CNCCs[30], was predicted as the true TG of a distal human biased enhancer, which was located near *HEPACAM* and far from *ROBO3*'s gene body (Fig. 2f). This distal human biased enhancer was validated by human-specific ATAC-seq and H3K27ac ChIP-seq signals and was consistent with the human specific expression pattern of *ROBO3* (FPKM 2.50 in human and 0.71 in chimpanzee, Fig. 2f). Though this human biased enhancer was nearest to *HEPACAM*, it was not associated with *HEPACAM* since *HEPACAM* was not expressed in human CNCC (FPKM 0.08). In addition, there were some CRM-TG regulations in hReg-CNCC that can be validated by Capture-C assay. For example, two REs were predicted by hReg-CNCC to regulate *SOX9*. One RE was located on *SOX9*'s promoter and the other RE was located at the 45k downstream of *SOX9*. It was noted that the distal RE and *SOX9* were linked by a loop of Capture-C data (Fig. 2g). As a comparison, the ABC model predicted 6 REs to regulate *SOX9* and only one of them can be validated by loops of Capture-C (Fig. 2g). This again shows the outperformance of hReg-CNCC to predict CRM-TG regulation.

Taken together, these results showed that our hReg-CNCC was capable to provide high-quality TF-TG, TF-CRM, and CRM-TG regulations.

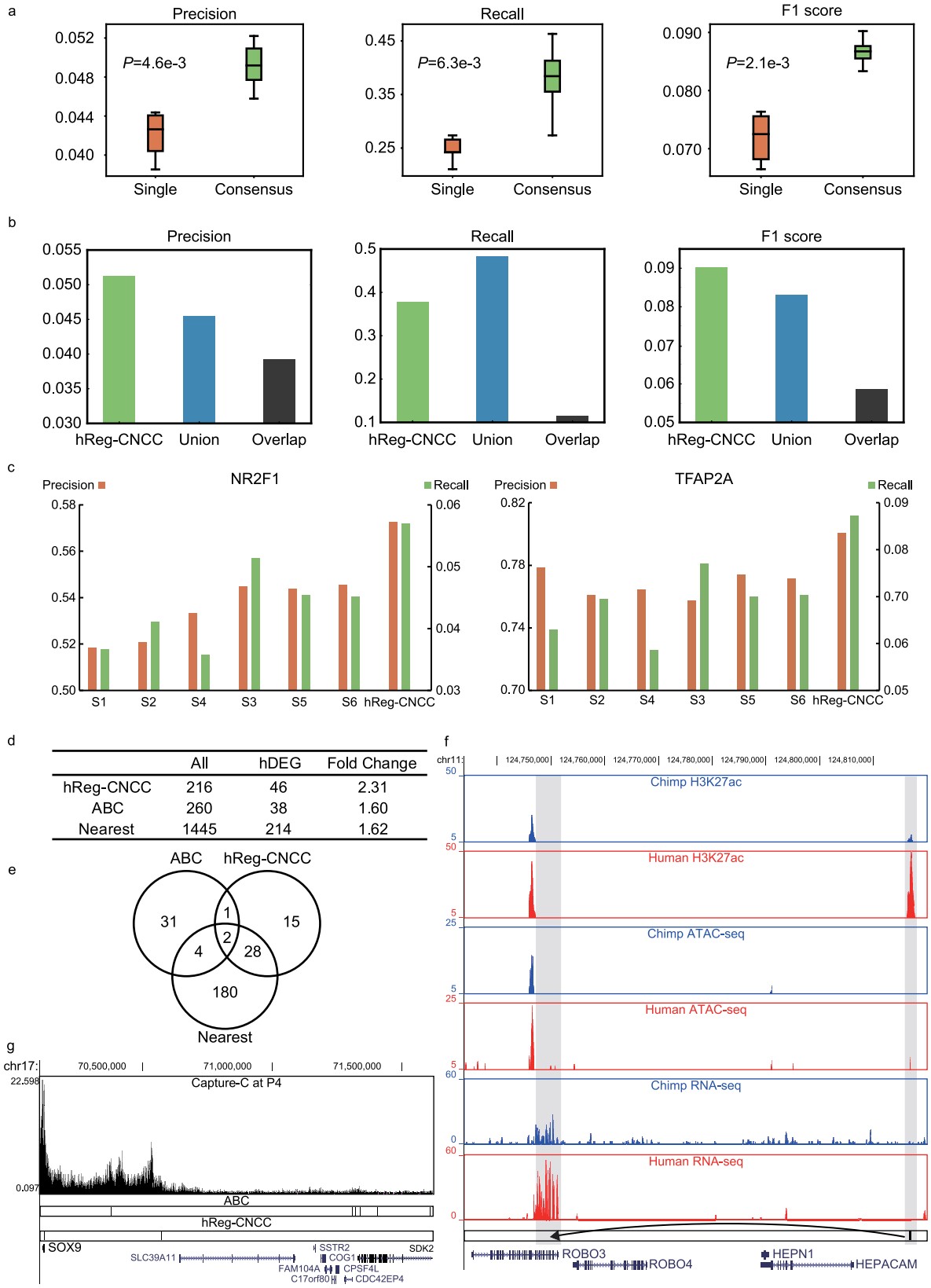

**hReg-CNCC reveals CNCC's regulatory architecture and core regulators.** After establishing that hReg-CNCC provided high-quality TF-TG, TF-CRM, and CRM-TG regulations, we next dissected the regulatory structure of hReg-CNCC and associated with CNCC's functions. We firstly decomposed hReg-CNCC into modules by hierarchical clustering on the inferred TF-TG

regulatory strength matrix. Two main regulatory modules can be detected from hReg-CNCC and were visualized in heatmap (Fig. 3a). We confirmed that the clustering was not driven by motif's trivial information content (Supplementary Fig. 2a, b). Module 1 was composed of 32 TFs (Table 1) and 7270 genes. The heatmap showed that those 32 TFs regulated almost all the TGs

**Fig. 2 Validating hReg-CNCC by independent data sources show that consensus optimization outperforms the alternative methods. a** Consensus optimization achieves significantly higher precision, recall, and F1 measure than single networks. The improvement is robust to parameter choices. $N = 6$ for single networks and $N = 6$ for consensus optimization. One-tailed $T$-test is conducted to obtain $P$-values. **b** Consensus optimization outperforms the naive union and intersection methods in precision, recall, and F1 measure. **c** hReg-CNCC can better predict two master regulators' (*TFAP2A* and *NR2F1*) ChIP-seq binding sites than single networks. **d** Using human biased differentially expressed genes as the gold standard, hReg-CNCC predicts the human biased enhancers' target genes more accurately than ABC model and proximity-based method. **e** hReg-CNCC predicts 33% enhancer gene relationships as distal regulation for human biased enhancers, which cannot be found by ABC model or proximity-based method. **f** RNA-seq, ATAC-seq, H3K27ac ChIP-seq track around *ROBO3*. hReg-CNCC predicts *ROBO3* as the target gene for a distal human biased enhancer (comparing human and chimpanzee's ATAC-seq tracks and H3K27ac tracks), which is located near the gene body of *HEPACAM*. The expression pattern of *ROBO3* supports the target assignment for the human biased enhancer (comparing human and chimpanzee's RNA-seq tracks). **g** Capture-C and REs tracks around *SOX9*. Two REs are predicted by hReg-CNCC to regulate *SOX9*. One RE is on the promoter of *SOX9* and the distal RE is validated by a loop of Capture-C anchored by *SOX9*.

(Fig. 3a, Supplementary Fig. 2c). Some of these TFs were known key regulators in CNCC pathway, such as *TFAP2A/B* and *NR2F1/2*[21]. Other TFs in Module 1, such as ALX family, POU family, TCF family, and TEAD family proteins, shared a very similar regulatory pattern as *TFAP2A/B*, indicating that they may also be vital in CNCC development. Some TFs were not included in CNCC pathway but also important for CNCC development. We labeled them as "other CNCC TFs" (Table 1). For example, *ALX1* was a pivotal regulator of echinoderm skeletogenesis[31] and associated with traits in frontonasal face[32]. Module 2 was a smaller one with 71 TFs and 1,544 TGs. Functional enrichment analysis showed that TFs in Module 2 were associated with specific biological processes in CNCC development (Table 2, Supplementary Fig. 2d). For example, "Mesenchyme development" was linked to neural crest development; "Skeletal system development" was associated with the skeleton of the face and head. *SIX1* and *TWIST1* in Module 2 were included in this GO term and they were top markers in tissues of "skeletal muscle" and "mESC to chondrocyte differentiation day 7"[33]; "Sensory organ development" was associated with sensory neurons in the peripheral nervous system and *LEF1* in this term was associated with "thalamus subthalamic nucleus"[33]; "Muscle structure development" and "Appendage morphogenesis" may be involved with muscle organization and ear development; Module 2 were also enriched in "heart development" and *HEY2* in module 2 was one of the markers in "heart atrium"[33]. From these observations, we hypothesized that Module 2 were responsible for developmental and differentiation functions and those TFs' regulations were involved in a more specific process of CNCC (Supplementary Fig. 2c). For example, *SIX1* and *SIX2* played a crucial neural crest cell-autonomous role in frontonasal morphogenesis[34]; *ZIC1* was responsible for triggering the early neural crest gene regulatory network by direct activation of multiple key neural crest specifiers[35], such as *SNAI1/2*, *FOXD3*, and *TWIST1*.

We further examined the relationships between the TFs in Module 1 and TFs in Module 2. Those TFs served as the backbone of our hReg-CNCC network and their regulation will reveal the gene regulation architecture in CNCC. We extracted the subnetwork with those TFs and associated CRMs. First, we obtained a TF regulatory subnetwork, defined as the directed graph with 103 TFs in the two modules as nodes and 10,609 TF-TF regulatory strength as edge weights. Then we extracted its backbone as the dense subnetwork with the maximal node and edge weights by quadratic programming (Methods). The extracted subnetwork was significantly denser than those randomly obtained from hReg-CNCC networks with the same number of nodes ($t$-test $P$-value $2.79 \times 10^{-7}$). The dense subnetwork was further partitioned into (i) the core subnetwork consisting of TFs that densely cross-regulated each other to achieve robust maintenance of the cellular state, (ii) the upstream subnetwork consisting of TFs that may regulate the core, and (iii) the downstream subnetwork consisting of key TFs regulated by the upstream and core subnetworks (Methods). The upstream

and core TFs were mainly the known master regulators such as *TFAP2A* and *NR2F1* (Fig. 3b). The downstream TFs contained many CNCC developmental TFs, such as *PAX3*[35] and *SIX1*[34]. We found that this dense network nicely connected the two modules in the regulatory structure of hReg-CNCC: all upstream TFs and core TFs were included in Module 1 (hypergeometric test $P$-value $< 1.09 \times 10^{-18}$) and 60% downstream TFs were contained in Module 2 (hypergeometric test $P$-value $< 4.49 \times 10^{-8}$). Heat-map of TF-TF regulatory strength revealed that TFs in Module 2 (downstream TF) were regulated by TFs in Module 1 (upstream and core TF) (Fig. 3c). Thus, our hReg-CNCC network structure analysis revealed regulatory hierarchy in CNCC, i.e., Module 1 contains TFs at higher network level that broadly regulated TFs in Module 2 as well as further downstream TGs (Supplementary Fig. 2c). Without using the module structure in hReg-CNCC, we used TF sub-network of hReg-CNCC to extract the dense subnetwork and reached very similar results (Supplementary Fig. 2e).

The regulatory hierarchy of TFs in two modules was consistent with known CNCC pathways[17] (Fig. 3d). *TFAP2A* and *TFAP2B* were the most upstream regulators and affected all other TFs in the CNCC dense network. Consistently, they were associated with all three CNCC functions. In contrast, the TFs of CNCC pathways in Module 2, which were mostly downstream TFs in the hierarchy, were involved in more specific functions. For example, *SMAD3* was associated with "neural plate border"; *SIX1* was associated with "neural crest specification"; *ZIC1*, *MSX1*, *MYC* were linked to "neural plate border" and "neural crest specification"; and *ETS1*, *SOX9* were linked to "neural crest specification" and "neural crest migration".

To evaluate the reproducibility of hReg-CNCC and its hierarchical architecture, we built another regulatory network (hReg-CNCC-H9) with an independent CNCC dataset[36]. hReg-CNCC-H9 was based on the paired RNA-seq and ATAC-seq data of human H9-ESC differentiated CNCC dataset[36] and was reconstructed with the same consensus optimization model of hReg-CNCC. First, we found significant overlapping of TFs, TGs, REs, and TF-TG regulations between hReg-CNCC and hReg-CNCC-H9 (Fig. 3e), revealing these genes and REs were indeed active in CNCC context. Second, we found that there were also two modules in hReg-CNCC-H9 (Fig. 3f): TFs in the first module broadly regulated most of the TGs and were significantly shared with Module 1 TF of hReg-CNCC (Fig. 3g, $P$-value $\leq 5.11 \times 10^{-22}$); the regulations of TFs in the second module were much more specific and was significantly overlapped with Module 2 TFs in hReg-CNCC (Fig. 3g, $P$-value $\leq 1.65 \times 10^{-40}$). This indicated that the two-module architecture of hReg-CNCC was reproducible. Third, we obtained the dense TF network of hReg-CNCC-H9 as we did for hReg-CNCC. We observed a consistent hierarchy of TFs between hReg-CNCC and hReg-CNCC-H9 (Supplementary Fig. 2f). For example, the upstream and core TFs, which were at a higher level of the regulatory network, were largely shared,

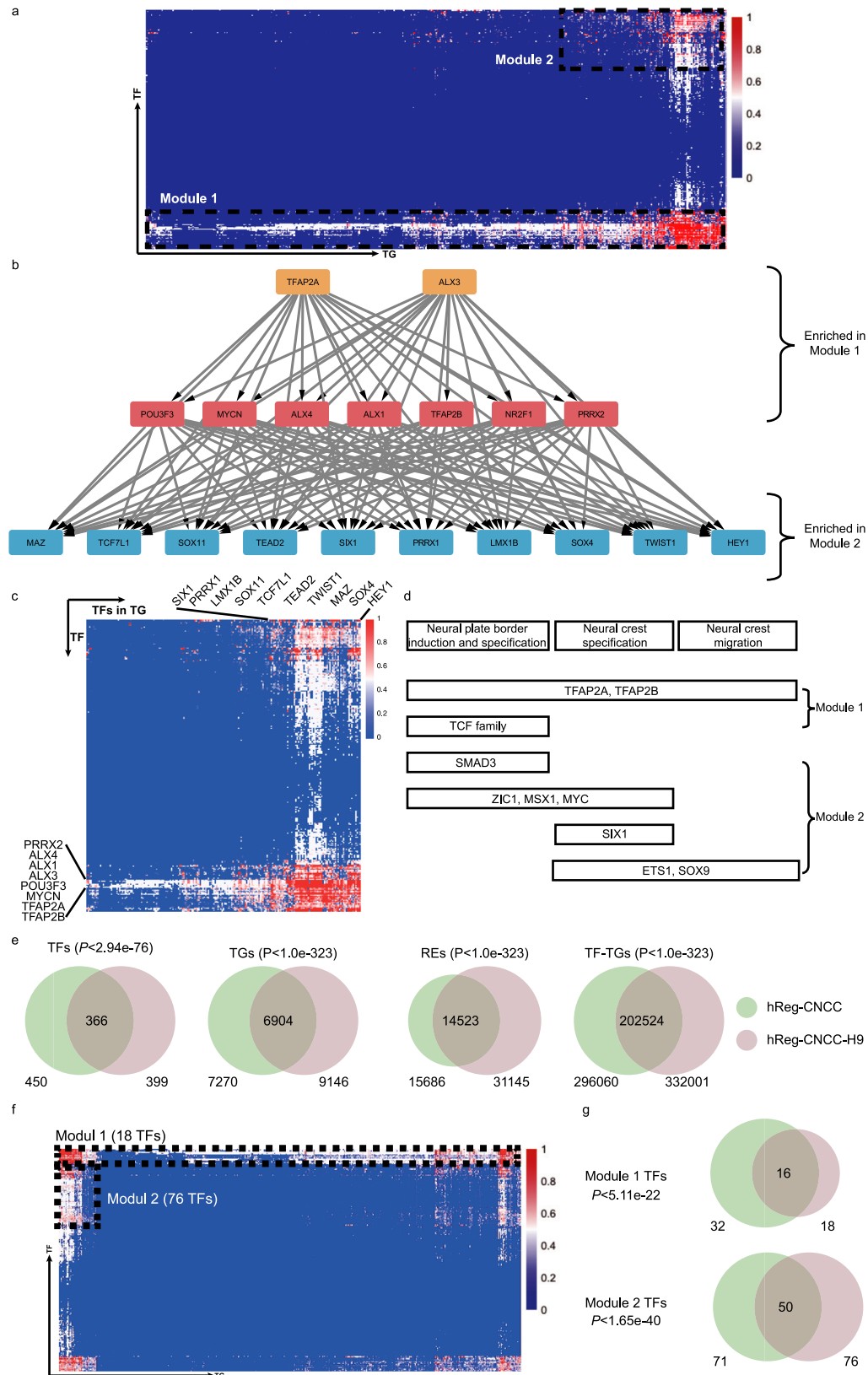

including *TFAP2A/B*, *ALX1/3/4*, *NR2F1*, *PRRX2*, and *MYCN*. And the downstream TFs of hReg-CNCC-H9 and hReg-CNCC were also overlapped, such as *TWIST1*, *SIX1*, *TCF7L1*, *LMX1B*, and *SOX4*. It was noted that hReg-CNCC and hReg-CNCC-H9 were significantly but not fully overlapped, which may result from the different biological material they used (iPSC for hReg-CNCC,

hESC for hReg-CNCC-H9). These results showed that the hReg-CNCC and its hierarchical architecture was well-validated and revealed the biological property of CNCC.

Collectively, hReg-CNCC revealed the hierarchical regulatory architecture of CNCC. There were two main TF-TG regulatory modules. TFs in Module 1, such as *TFAP2A* and *NR2F1*[37], were at

**Fig. 3 Architecture of hReg-CNCC reveals the regulatory hierarchy in CNCC. a** Heatmap of hReg-CNCC's regulatory strength adjacent matrix shows two TF-TG modules with different regulatory patterns. X-axis denotes TG and y-axis is TF. TFs in Module 1 tend to regulate a large number of TGs. TFs in Module 2 specifically regulate a subset of TGs. **b** Dense network extracted from hReg-CNCC for the TFs in Module 1 and Module 2 shows a clear hierarchical structure. TFs in Module 1 tend to be upstream and core regulators. TFs in Module 2 tend to be downstream TFs associated with CNCC's specific development, differentiation, and migration. The hierarchy is consistent with GO function enrichment results. **c** Heatmap of TF-TF regulatory strength matrix shows a consistent two-module structure of hReg-CNCC. **d** Overlapping TFs in Module 1 and Module 2 with the known CNCC pathways. Hypergeometric test is conducted to obtain the P-values. **e** Overlap of TFs, TGs, REs, and TF-TG regulations between hReg-CNCC-H9 and hReg-CNCC. **f** Heatmap of hReg-CNCC-H9 reveals two-module architecture. **g** The TFs of the two modules are significantly shared by hReg-CNCC-H9 and hReg-CNCC. A hypergeometric test is conducted to obtain the P-values.

**Table 1 TFs in Module 1 are annotated as CNCC markers, pathways and other CNCC TFs.**

| TFs in Module 1 | | | | Annotation |
|---|---|---|---|---|
| TFAP2A | TFAP2B | NR2F1 | NR2F2 | CNCC markers |
| TFAP2A/B | TCF4 | TCF3 | TCF12 | CNCC pathway |
| PRRX2 | ALX4 | ALX1 | ALX3 | Other CNCC TFs |
| MYCN | POU3F3 | POU3F1 | POU2F1 | |
| ERG | FLI1 | SP2 | ETV4 | |
| CTCF | E2F2 | HEYL | HEY1 | |
| RXRA | TEAD4 | TEAD2 | TEAD3 | |
| SOX11 | PRRX1 | SHOX2 | PHOX2A | |
| RXRB | | | | |

a higher level of the regulatory network and broadly regulated other genes. TFs in Module 2 were at a lower level and largely regulated by TFs in Module 1. They specifically regulated CNCC's functions of induction, specification, and migration.

**Annotating SNPs of human facial variation with hReg-CNCC.** The regulations in hReg-CNCC inferred from accessibility and expression data may provide tools for interpretation of genetic variants. The genetic variants in the CRMs of hReg-CNCC, including their functional REs, TGs, and bound TFs, should be useful in the annotation of SNPs identified by GWAS of human facial variation traits. To demonstrate this, we extracted the SNPs with their summary statistics from GWAS study on 78 distance phenotypes among 13 landmarks in human face[2]. In total there were 495 significant SNPs with P-value $< 5 \times 10^{-8}$ and on average there were 7 SNPs for each distance phenotype. We firstly calculated the fold change enrichment of facial shape-associated SNPs in hReg-CNCC's REs, in CNCC's ATAC-seq peaks, and in other tissues' peaks as control (Methods). First, we observed FC score of all three region sets decreased when threshold P-value $\leq 1 \times 10^{-6}$ and this may result from the insufficient number of SNPs (Fig. 4a). There were only 1762 SNP with threshold P-value $\leq 1 \times 10^{-6}$. This motivated us to focus the enrichment analysis on SNPs within P-value $\leq 1 \times 10^{-5}$. The results showed that those facial shape-associated SNPs were much more enriched in hReg-CNCC REs with fold enrichment 1.3 when P-value cutoff is $1 \times 10^{-5}$ (Fig. 4a). Facial shape-associated SNPs have moderate enrichment in CNCC peaks and no enrichment in other tissues (Fig. 4a). And the Facial SNPs' enrichment in hReg-NCCC were higher than the random SNP set generated by SNPsnap[38] (Supplementary Fig. 3). This again demonstrated that hReg-CNCC greatly improved the CNCC REs' quality by integrating TF/TG expression and enhanced replicate reproducibility by consensus optimization.

With the global enrichment signal, we next scanned every facial shape-associated SNPs with P-value $\leq 1 \times 10^{-5}$ in TF-CRM-TG triplet of hReg-CNCC (no SNPs with P-value $\leq 5 \times 10^{-8}$ were overlapped with hReg-CNCC). For each distance phenotype, if its

SNP was located in one CRM, we linked the SNP to this TF-CRM-TG triplet. Then we pooled all the SNP-triplet links together and formed a SNP associated regulatory sub-network associated with this distance phenotype (Fig. 4b). Taking all 78 distance phenotypes associated sub-networks together, we can get the SNPs associated network for all face distance phenotypes (Methods, Fig. 4c). This subnetwork allowed us to explore the relationships between SNPs, regulations, and phenotypes.

In total 22 SNPs and 28 TGs were identified. The region of the nose and mouth included more genes (15/28) and SNPs (14/22) than other regions, which was consistent with the previous findings that it was the most heritable region[39,40]. Among these SNP associated TGs, there were 6 TFs: *ALX1*, *CNOT3*, *MLLT1*, *POLR2J*, *RUNX2*, *SFPQ* and the other 16 genes were downstream TGs. Literature evidence supported that mis-regulation of these TFs was involved with a facial abnormality. For example, *CNOT3* was associated with IDDSADF (an intellectual developmental disorder with speech delay, autism, and dysmorphic facies), which will cause abnormal facial morphosis[41]. *SFPQ* formed a complex with *TFII-I* and *PARP1* and regulated *DYX1C1* implicated in neuronal migration and dyslexia[42]. *RUNX2* was known as a master TF of bone and played a role in the development of the teeth and supporting structures[43], which was associated with the mouth region. And the *SUPT3H-RUNX2* locus was reported by previous GWAS to be associated with bone and cartilage phenotypes[44]. Among the 6 TFs, *ALX1*, and *RUNX2* were involved in regulating other genes in hReg-CNCC. There were also many SNPs and downstream TGs, such as *RUVBL1*[45], which was also associated with the nose and mouth.

In the face SNP associated subnetwork, we found that even though different traits at a different region of the face had different SNPs and TGs, they shared a group of upstream TFs. For example, *TFAP2A* regulated 19/28 of the TGs; *NR2F1* regulated 16/28 of the TGs, and *ALX3/4* regulated 22/28 of the TGs. These TFs were in Module 1 and upstream or core TFs in dense TF network of hReg-CNCC, which was consistent with the CNCC's regulatory architecture that TFs in Module 1 broadly regulated other genes.

To find out the key CRMs in face SNP associated subnetwork, we ranked the TF-CRM-TG triplets by their regulatory strength (Methods) and found *ALX1* was regulated by several high-ranking CRMs (Fig. 4d). We noticed that *ALX1* was the only one Module 1 TF in face SNP associated subnetwork. In addition, *ALX1* was the core TF in a dense TF network and regulates many Module 2 TFs. Together these evidences suggested *ALX1* as the candidate facial shape-associated gene in our annotated regulatory network. There were two SNPs (rs12810608, rs11609649) associated with *ALX1*. SNP rs12810608 (P-value 3.30e−07) was located in *ALX1*'s promoter and SNP rs11609649 (P-value 1.55e−06) was located in a distal regulatory region (97K upstream). The promoter and distal RE were supported by both ATAC-seq and H3K27ac signals (Fig. 4e). Even though the distal RE and SNP were in *LRRIQ1*'s gene body, they were not associated with *LRRIQ1* since it was nearly not expressed in CNCC (FPKM 0.49).

**Table 2 TFs in Module 2 are enriched in CNCC-specific developmental terms.**

| Function | −log(P-value) | Associated TFs in Module 2 |
|---|---|---|
| Chordate embryonic development | 29.93 | GABPA,LMX1B,SMAD3,MSX1,SIX1,SOX9,SP3,SRF,TCF7,TWIST1,ARNT2,SIX2,HEY2,LEF1,SIX4,ETS2, **MSX2**,ZIC1,KLF4,EMX2,RFX3,BARX1 |
| Mesenchyme development | 21.28 | SMAD3,MSX1,MSX2,MYC,SIX1,SOX9,TWIST1,SIX2,HEY2,LEF1,SIX4,FOXO4,NFE2L2,SRF,POU6F2, TCF7L2 |
| Skeletal system development | 21.18 | ESRRA,ETS2,SMAD3,MSX1,MSX2,**SIX1**,SOX9,SP3,SRF,**TWIST1**,SIX2,SIX4,EGR1 |
| Tissue morphogenesis | 18.91 | ETV5,SMAD3,MSX1,MSX2,MYC,SIX1,SOX9,SRF,TWIST1,KLF4,SIX2,HEY2,LEF1,SIX4,EGR1,EMX2 |
| Response to growth factor | 18.49 | E2F1,EGR1,ELK1,FOXO3,SMAD3,MSX1,MSX2,MYC,SOX9,TWIST1,KLF4,LEF1,ZBTB7A,FOXO4 |
| Organ sensory development | 26.85 | SMAD3,MSX1,SIX1,SOX9,SP3,SRF,TWIST1,ZIC1,KLF4,SIX2,HEY2,**LEF1**,SIX4 |
| Vasculature development | 14.86 | EGR1,ELK3,ETS1,FOXO4,NFE2L2,SIX1,SP1,SRF,TCF7L2,TWIST1,KLF4,HEY2,LEF1,FOXJ2,SMAD3,MYC, SOX9,SIX4,MAZ,TWIST2 |
| Ossification | 14.18 | ESRRA,SMAD3,MSX2,SOX9,SP3,**TWIST1**,SIX2,LEF1,**TWIST2**,HEY2 |
| Regulation of animal organ morphogenesis | 13.78 | ETV5,MSX1,MYC,SIX1,SOX9,TWIST1,SIX2,SIX4,EGR1,FOXO3,SMAD3,TCF7,TCF7L2,KLF4,LEF1,BARX1, TCF7L1 |
| Regulation of neuron differentiation | 12.72 | ETV5,FOXO3,NFE2L2,SIX1,SOX9,SRF,KLF4,HEY2,E2F1,**ETV1**,LEF1,RFX3 |
| Muscle structure development | 12.01 | EGR1,ETV1,SMAD3,FOXO4,MSX1,**SIX1**,SOX9,SRF,TCF7L2,TWIST1,HEY2,LEF1,SIX4 |
| Appendage morphogenesis | 11.87 | MSX1,MSX2,SOX9,**TWIST1**,LEF1 |
| Heart development | 11.84 | SMAD3,MSX1,MSX2,SIX1,SOX9,SRF,TWIST1,**HEY2** |
| Mesenchyme morphogenesis | 11.78 | SMAD3,MSX1,MSX2,MYC,SOX9,TWIST1,HEY2,LEF1 |

The genes with bold characters are the marker of the function in the first column.

And the chromatin accessibility of this distal RE had 0.51 Pearson correlation with *ALX1*'s expression across diverse samples (Supplementary Fig. 4a). The above evidence demonstrated that hReg-CNCC can accurately predict distal REs as the regulator of *ALX1*. Then we sought to find out which TFs' binding affinity were potentially influenced by these two SNPs. We scanned motif on these two REs with effective allele and reference allele of the SNPs respectively and found that the binding affinity of many motifs were changed (Fig. 4e). There was a gain of motif "PB0186.1_Tcf3_2/Jaspar" in promoter and its associated TCF clusters were top regulators of *ALX1* (Supplementary Fig. 4b). On the distal RE, when allele at rs11609649 was changed to the effective allele, there were a gain of motif "PH0082.1_Irx2/Jaspar" and a loss of motif FOXM1_1/encode. These two motifs corresponded to *IRX3* and *FOXM1* respectively, which were highly expressed in CNCC (FPKM of *IRX3* 68.12, *FOXM1* 49.03) and played roles in development of CNCC and derivatives[46,47] (Fig. 4e, Supplementary Data 2). Several lines of evidence supported that these two SNPs were respectively located in promoter and distal enhancer, potentially altered the regulation of *ALX1* by influencing TFs such as TCF cluster, *IRX3* and *FOXM1*, combinatorially tuned *ALX1*'s expression level, propagated the regulatory information in hReg-CNCC network, and led to the phenotype changes.

hReg-CNCC provided a means to extract regulatory relations among SNPs, REs, TGs, and phenotypes. We revealed several interesting patterns to potentially illustrate SNPs' multi-trait effect and cooperation. There were two patterns of multi-trait effect: the first was that one SNP was located in a RE and this RE regulated multiple genes. For example, rs16985457 was located in chr19:54693360-54695240 and chr19:54693360-54695240 was predicted to regulate *CNOT3*, *PRPF31*, and *LENG1* (Fig. 4f); the second type was one SNP, which was associated with multiple traits, was located in a RE and this RE only regulates a gene. For example, rs12810608 was located in chr12:85673460-85674718, which regulates *ALX1*, and this SNP was associated with three face distances: "EnR-Prn", "EnL-Prn", and "EnR-AlL" (Fig. 4g). In addition, multiple SNPs cooperated in one RE and worked together to influence the activity of RE. For instance, rs11719548 and rs11711710 were simultaneously located in chr3:12872024-

127872868 and chr3:12872024-127872868 regulated *RUVBL1* (Fig. 4h).

In summary, hReg-CNCC can improve the enrichment of facial shape-associated SNPs in CNCC and used the TFs, REs, and TGs to help explain how genetic variants get involved in regulation, such as *ALX1*. hReg-CNCC can also illustrate the possible mechanism of SNPs' multi-trait effect and cooperation.

**hReg-CNCC holds the potential to enhance mechanism understanding for other facial traits.** Neural crest was a multipotent cell population that contributed to a wide variety of derivatives by the dedicate regulations among TF, CRM, and TGs. Possible defects in the regulation of CNCC development contributed to a large percentage of congenital birth defects. This fact motivated us to depict human facial traits' the known genetic variants by its TGs and regulation. In GWAS Catalog, there were a total of 18 traits/diseases that were related to human facial variation and we collected their significant SNPs. We found that 1/3 of these face associated traits' significant SNPs were associated with CRMs in hReg-CNCC (Fig. 5a). For instance, one significant SNP rs758468472 was reported to be associated with "Tooth agenesis" (GWAS with 340,498 participants and 9 risk variants[48]) in GWAS catalog. hReg-CNCC showed that it located in "chr17:65713670-65714238" and this RE was predicted to regulate *NOL11* (Fig. 5a), which played major roles in CNCC development[49]. One of the significant SNPs of "Monobrow" (GWAS with 69,000 participants and 61 significant loci[50]) rs11609649 was located in chr12:85576368-85576871, whose TG was *ALX1* and binding TFs were ALX cluster, *TFAP2B/A*, *POU3F3*, *PRRX2*, and *NR2F1* (Fig. 5a). Interestingly, rs11609649 and *ALX1* were also detected in annotating GWAS of face distances and their related phenotypes were frontonasal distances (Fig. 5b), indicating that these two traits may be correlated. These results agreed with the fact that *ALX1* was associated with frontonasal face[32] since "Monobrow" and frontonasal distances were both characteristics of frontonasal face. Putting all the evidence together, SNP rs11609649 was located in a distal RE (97K upstream, chr12:85576368-85576871) and regulated *ALX1*. This regulation may be influenced by SNP rs11609649 on the motif

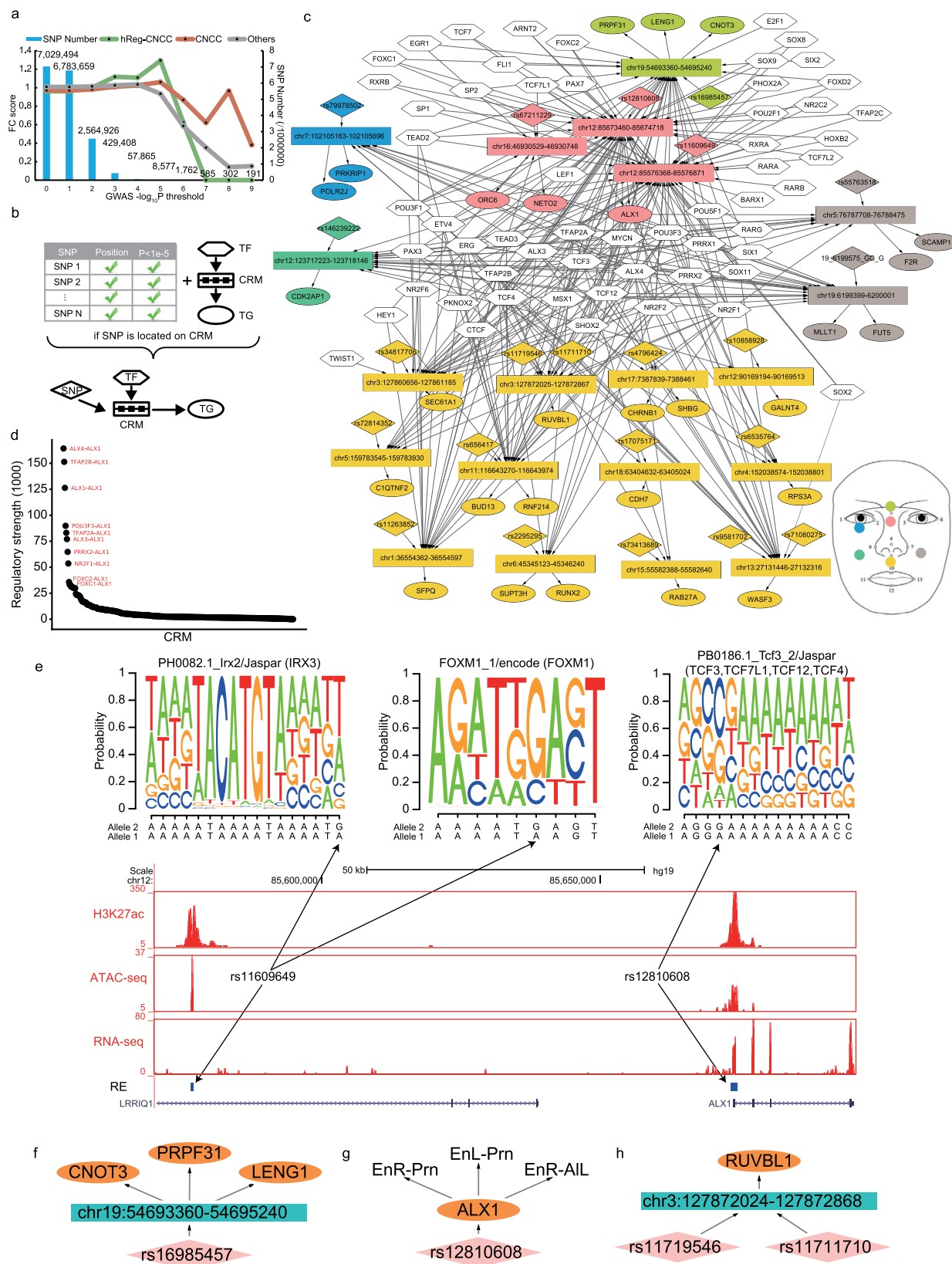

binding of *IRF3* and *FOXM1* (Fig. 4d). This mechanism was probably responsible for both normal regulations of frontonasal distances and abnormal phenotype "Monobrow" (Fig. 5c).

On the other hand, with known causal genes of rare diseases manifesting facial abnormalities, we can use hReg-CNCC to reveal their REs and regulating TFs. For example, Williams-Beuren Syndrome (WBS), a disease with the characteristic of the milder face, had a causal gene *BAZ1B*, found by sequence deletion of WBS patients[51]. We used hReg-CNCC to study *BAZ1B* and found there were five REs composed its upstream CRM (Fig. 5d).

**Fig. 4 hReg-CNCC identifies causal regulations for genetic variants and reveals biological insights for genotype and phenotype mapping. a** Face GWAS SNPs are more enriched in CRMs in hReg-CNCC than CNCC ATAC-seq peaks by fold change along with –log(P-value). Other tissues' fold change is the mean of 27 samples (Supplementary Data 4). **b** Procedure to associate TF-CRM-TG triplet with significant SNPs passing threshold. If one SNP is located on CRM, then the TF-CRM-TG is linked by this SNP and form a SNP associated TF-CRM-TG. **c** The face SNPs associated TF-CRM-TG network. In the network, REs associated with SNPs are shown instead of the whole CRM. The colors of SNP, RE, and TG indicate different sub-phenotypes for the face illustrated in the right corner. **d** Ranking the regulatory strength in the face SNP associated network shows that *ALX1* is a pivotal regulator. **e** RNA-seq, ATAC-seq, H3K27ac, and RE tracks around *ALX1*. Two SNPs are located in REs of *ALX1*: one in promoter, the other in 97K upstream cis-regulatory region. The SNP in promoter changes the binding affinity of *TCF* cluster and the SNP in distal RE changes the binding affinity of *IRX3* and *FOXM1*, which causally alter *ALX1*'s expression level and further face phenotype is given that *ALX1* is the master regulator in CNCC's migration. **f, g** Examples of two types of SNPs' multi-trait effect in SNPs associated network. **h** Multiple SNP cooperation example in SNPs associated network. EnR Right Endocanthion, EnL Left Endocanthion, Prn Pronasale, AlL Left Alare.

Two REs were near promoter or in the gene body and the other three were distal REs. Among its upstream TFs, *TFAP2B/A* and *MYCN* regulated *BAZ1B* through all five REs; *ALX4* bound on two of these REs. Two of these REs were linked to some SNPs of GWAS of 78 distance[2]. One was located in promoter and the other was in a cis-RE. These SNPs' most associated distance traits (defined as the SNP's minimal P-value corresponded distance trait) were face width (Fig. 5d), which was probably linked to characters of WBS: puffiness around the eyes and full cheeks. Taken together, *BAZ1B* was regulated by a CRM consisting of five REs, bound by TFs such as *TFAP2B/A*, *MYCN*, and *ALX4*. These TFs binding was probably affected by two SNPs, rs62466263 and rs73134905, and finally exerted influence on face width and led to milder face characteristics.

**Interpreting DNA difference in evolution with hReg-CNCC.** The evolutionary elements of the human genome contributed to many characteristics of human[52], including face morphosis. It was promising to annotate evolutionary elements with hReg-CNCC to reveal their regulation on the face. Similar to the SNP annotation procedure, we checked every TF-CRM-TG triplet and if its CRM was overlapped with some human evolutionarily important elements, it was extracted to form the annotated network (Fig. 6a).

We first focused on the 481 human ultra-conserved elements (UCEs)[53], which were identical in at least three of five placental mammals (human, dog, cow, mouse, and rat). We found five REs in hReg-CNCC were overlapped with human UCEs (P-value < 0.0074, Method, Fig. 6b). Among them, three elements were also candidate enhancers in VISTA database[54]. And one of these three VISTA enhancers "chr8:77690693-77691421" was positive for transgenic mouse assay (4/4 were limb positive and 1/4 was neural tube positive), but showing no activity in the developing facial structures at E11.5 of the mouse embryo. Its downstream TG was *ZFHX4*, which was a known causal gene for "Congenital Ptosis"[55]. The other two vista enhancers were also associated with facial traits. For example, "chr5:77148376-77148723" was linked to *TBCA*, a causal gene for HRDS (Hypoparathyroidism-Retardation-Dysmorphism Syndrome)[56], which will cause facial anomalies; the downstream gene of "chr1:244217544-244217918" was *ZBTB18* and it was associated with MRD22[57], whose symptom included variable but characteristic facial features. In the UCE associated subnetwork, we noticed that there were two types of TFs that were consistent with hReg-CNCC revealed 2-Module regulatory architecture. For example, *TFAP2B*, *ALX4*, and *TCF4* represented the first type of TFs and they regulated four of the five annotated UCEs. Their property of broad regulation in UCE network agreed with the fact that they were in Module 1 and upstream or core TFs in a dense network. On the other hand, *TWIST1* and *TFAP4* represented the second type TFs and they only regulated one of the five annotated UCEs, showing their feature of specific regulation. This was in

accordance with the fact that they were in Module 2 or downstream TFs in a dense network. These results showed that hReg-CNCC helped find the conserved elements that were responsible for facial development and illustrated their regulations.

Next, we sought to annotate human accelerated regions[58] (HAR) which were conserved in other species but had dramatically increased substitution rates in the human lineage. In total, 13 REs in hReg-CNCC were found to be associated with HAR (P-value < 0.1001, Supplementary Fig. 5a). These REs were associated with the development of neural crest and facial diseases. For example, chr7:134379901-134380671 was predicted to regulate *CALD1*. *CALD1* was known to play an essential role in the regulation of smooth muscle and non-muscle contraction[59], which was a derivative of neural crest[60]. chr17:597438-597652 regulated *FAM57A*, which was linked to "Sclerosteosis 1", a disease with abnormal character of skull and mandible[61]. And the downstream TG of chr12:12472818-12473034 was *LRP6*, which was associated with "tooth agenesis"[62]. To understand the influence of these 13 REs more precisely, we overlapped them with SNPs of 78 face distances (Supplementary Fig. 5b). There were 9 SNPs involved with 5 of these 13 REs. Among these SNPs, 8 SNPs' most relevant face distances (defined as the SNP's minimal P-value corresponded distance trait) were associated with nose (Supplementary Fig. 5c), specifically "Pronasale", "Subnasale", "Left Alare", and "Right Alare". This area was the most heritable and explained most of the variance of human faces[40]. Together with its association with HARs revealed above, the nose area may also be the main difference between faces of human and other species. Besides the sequence difference, the accessibility of these 13 REs was different from chimpanzee with fold change range from 3.3 to 647.3 (Supplementary Fig. 5d). The downstream genes of these REs also showed a difference in human and chimpanzee. For example, *CHRNA3* had significantly different expression levels in human and chimpanzee and it was a trigeminal ganglion marker[63].

Together, hReg-CNCC can be used to annotate evolutionarily important elements of the human genome and illustrated their regulatory mechanism on face development.

## Discussion

In this paper, we proposed to construct a regulatory network in a developmental context to understand the genetic variants and complex phenotypes. CNCC was chosen as an embryonic cell population that gives rise to a multitude of derivatives. Paired gene expression & chromatin accessibility data were integrated to reconstruct the genome-wide human Regulatory network of Cranial Neural Crest Cells (hReg-CNCC). hReg-CNCC provided a valuable resource to interpret genetic variant as early as gastrulation during embryonic development. Importantly the architecture that upstream, core, and downstream TFs were associated with functions of neural plate border, specification, and

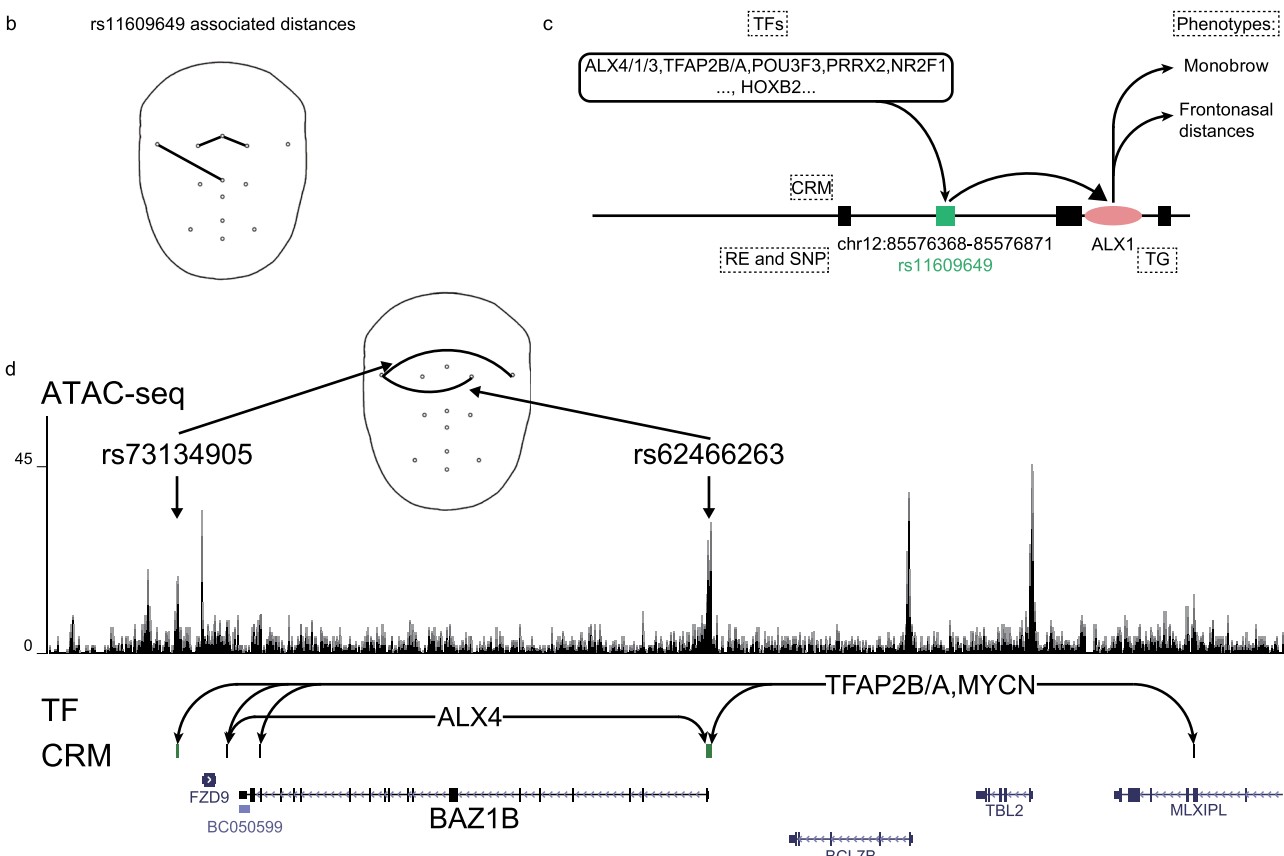

**Fig. 5 hReg-CNCC provides mechanism understanding for human face related diseases. a** 18 face associated traits in GWAS catalog are scanned and 6 (1/3) can be explained by hReg-CNCC with associated RE and TF-TG regulation. TFs were filtered to select top TFs (details in "Methods"). **b** rs11609609 is associated with frontonasal distances. **c** Detailed regulation of rs11609609 and *ALX1*, which influences frontonasal face and "Monobrow". **d** The upstream TFs and CRM of *BAZ1B* in hReg-CNCC support *BAZ1B* as a causal gene associated with the rare disease Williams-Beuren Syndrome. Two SNPs, rs73134905 and rs62466263, locate in the downstream enhancer and promoter (two REs in the CRM) of *BAZ1B*. These two SNPs are most associated with face width phenotype in GWAS study. Face width phenotype is consistent with wilder face symptom of WBS patients.

migration were explored in hReg-CNCC and biological insights were derived on how this architecture was associated with genetic variants of human facial GWAS, disease traits, and DNA sequence differences in evolution.

The human face is an exemplar complex morphological structure resulting from the intricate coordination of genetic, cellular, and environmental factors[64]. The scientific questions we are interested in is how it arises during development and evolved during evolution. One solid starting point is the genetic architecture of the human face characterized by GWAS to multivariate shape phenotypes. The traditional method to annotate GWAS variant is to use FUMA[65] or GREAT[66] to establish the association with genes located within 500 kb of the SNPs. Beyond this tissue non-specific and noisy annotation, the activity of GWAS regions can be assessed by the epigenomic mapping datasets. For example, ChIP-seq signals of H3K27ac is used as a marker of the

promoter of transcriptionally active genes and active distal enhancers. Here we propose a developmental context-specific network as one step further to annotate genetic variant by providing better resolution at <1 kb regulatory regions and detailed interpretation with upstream binding TFs and downstream TGs.

Reproducibility is essential to reliable scientific discovery in high throughput experiments. In this work, we propose a consensus optimization approach to measure the reproducibility of regulations identified from replicate experiments. The reproducibility concept was previously used in the IDR method for studying protein-binding regions on the genome by ChIP-seq assay[67]. We show consensus optimization outperforms the naive integrative methods and can fully utilize the information in biological replicates.

We note that the insufficient knowledge of human CNCC hinders the validation of hReg-CNCC. To battle the scarcity of

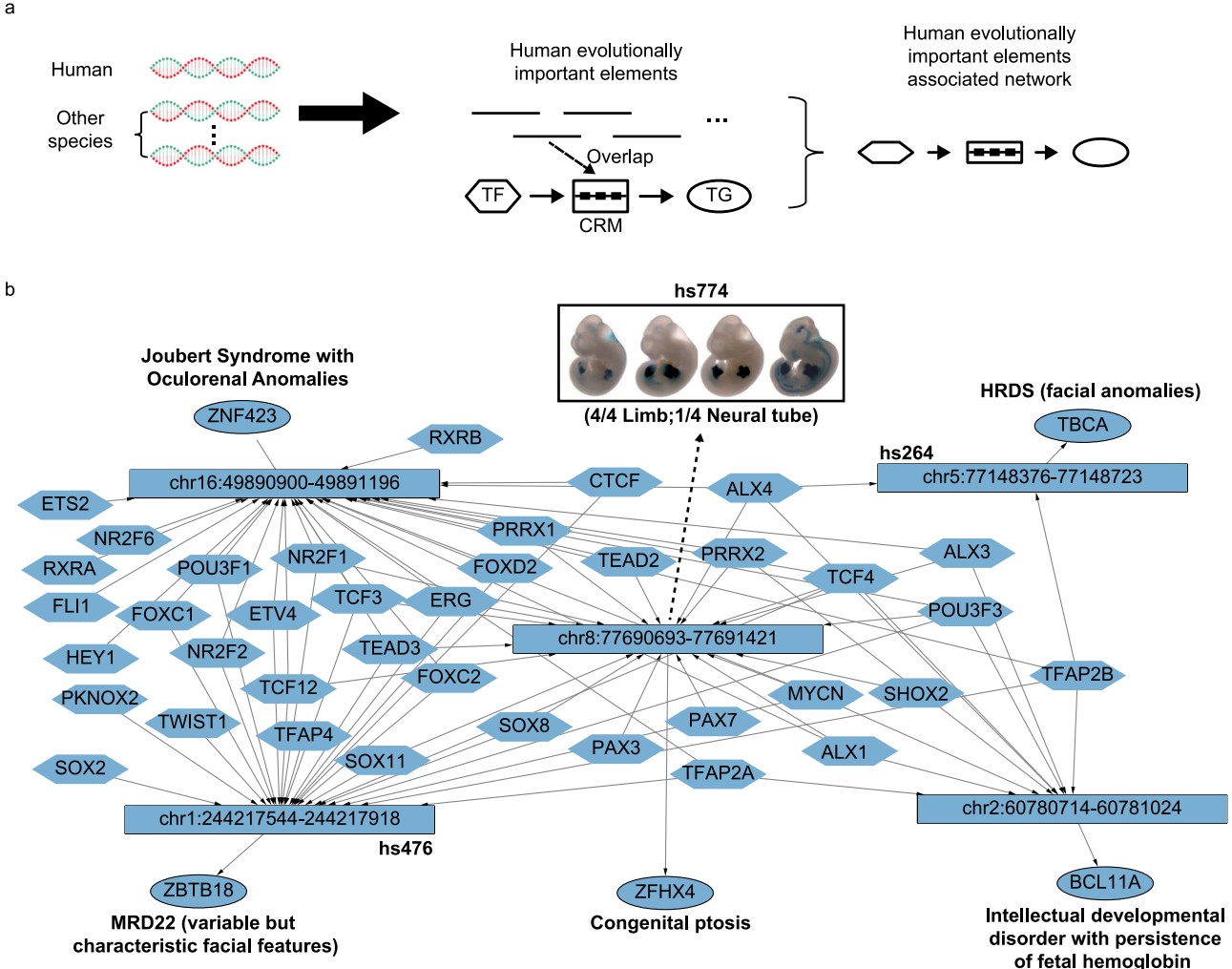

**Fig. 6 hReg-CNCC interprets the DNA difference in evolution and uncovers important regulatory elements and genes. a** The scheme to extract the subnetwork in hReg-CNCC associated with human evolutionarily important elements from comparative genomics. If one human evolutionarily important element is overlapped with CRM, this TF-CRM-TG triplet is extracted and pooled into a subnetwork. **b** The evolutionarily UCEs associated network. Instead of the whole CRM, only the REs associated with evolutionary elements are shown. Vista enhancer and literature evidence are annotated and support their importance in face development.

direct gold-standard positives, we use three independent data sources to approximately validate hReg-CNCC including CNCC pathways in non-human organisms, ChIP-seq data of core regulators in human CNCC, and expression data of human CNCC. Combined evidences of these three aspects support that hReg-CNCC performs better than other methods. In addition, we utilize recall, precision, and F1 score metric to make a comparison with other methods. The "low" precision rate and F1 score is caused by the imbalance of the predicted set and validated set. For example, among the 50 genes of CNCC pathway, our hReg-CNCC predicts 703 regulations with good coverage and there are only 90 edges in CNCC pathways, which are likely to be only partially annotated.

The three-germ lineage model of cell type has limitations and the neural crest has long been argued as a fourth germ lineage[68]. CNCC represents an early time point in facial development with known neural crest master regulators, *Nr2f1*, *Nr2f2*, *Msx1*, *Msx2*, and *Tfap2a*[37,69]. Currently hReg-CNCC only provides the snapshot regulation at this single time point. It's not surprising that hReg-CNCC can only interpret limited number of SNPs and the overall enrichment is quite modest. The craniofacial tissues represent progressively later time points and the intermediate cell types during development and their regulatory networks should be reconstructed in future.

Another limitation of this study is the use of significant SNPs from GWAS study to study their influence on upstream TF binding. Since GWAS detects the tag SNPs, which are useful to identify the genomic regions on the chromosome but do not have a direct causal relation to the phenotype. Thus, our predictions on SNP's influenced upstream TF binding should be interpreted carefully since they are only markers on chromosomes. We expect future progress on this problem will come from fine mapping with WGS data[70].

hReg-CNCC infers the regulation of gene expression as the interaction of TFs with DNA regions with open chromatin structure in CNCC and correlation of gene expression and chromatin accessibility across ENCODE tissue samples[23]. Much deeper understanding can be revealed by 3D chromatin interaction data to provide physical enhancer-promoter interactions[71] and time course regulatory analysis, in which both gene expression and chromatin accessibility are measured at each developmental time point in a time course experiment[26]. Furthermore,

CNCC is known as a heterogeneous mixture of many cell types. It will be fruitful to infer the regulatory networks of the underlying cell types based on scATAC-seq and scRNA-seq data[72]. Another possible application is about the cancer of CNCC derivatives, such as skin. We found that most of the REs of hReg-CNCC were also accessible in "lower leg skin", but inaccessible in Skin Cutaneous Melanoma (Supplementary Fig. 6a). Many REs of CNCC regulators, such as *SOX9/10*, were also inactive in Skin Cutaneous Melanoma (Supplementary Fig. 6b). This observation indicates the potential role of hReg-CNCC to study cancer of CNCC derivatives.

## Methods

**Ethical statement**. The methods were performed in accordance with relevant guidelines and regulations and approved by Academy of Mathematics and Systems Science, Chinese Academy of Sciences.

**Constructing regulatory network from paired gene expression and chromatin accessibility data by PECA2**. We utilized PECA2 to infer genome-wide and context-specific regulatory networks based on gene expression and chromatin accessibility data in that context[26]. Given paired RNA-seq and ATAC-seq data for a single replicate, PECA2 hypothesized that TF regulated the downstream TG by binding to CRM (Fig. 1). The main idea of CRM was proposed to combine several REs bound by the same TF to regulate one common TG.

The regulatory strength of a TF on a TG was quantified by the trans-regulation score, which was calculated by integrating information from multiple REs that may mediate the activity of the TF to regulate the TG. A prior TF-TG correlation across external public data from ENCODE database was included in the trans-regulation score definition to distinguish the TFs sharing the same binding motif (i.e., TFs from the same family). Specifically, a TF regulated a TG in CNCC if (1) the TF and TG were expressed in CNCC, (2) this TF's motifs were enriched in the REs of TG in CNCC, and (3) the expression of TF was highly correlated with this TG across diverse ENCODE samples. Taking these three factors into account, the trans-regulation score $S_{ij}$ of $i$-th TF and $j$-th TG was quantified as

$$S_{ij} = \left( \sum_k B_{ik} RE_k I_{kj} \right) \times 2^{|R_{ij}|} \times \sqrt{TFA_i TG_j} \quad (1)$$

Here $TFA_i$ represented the activity of the $i$-th TF and was calculated as the geometric mean of normalized expression $TF_i$ and motif enrichment score on open regions of CNCC. $TG_j$ was normalized expression of the $j$-th TG in CNCC. $B_{ik}$ was motif binding strength of $i$-th TF on $k$-th RE, which was defined as the sum of binding strength (motif position weight matrix-based log-odds probabilities given by HOMER) of all of the binding sites of $i$-th TF on this RE. $RE_k$ was the measure of normalized accessibility for $k$-th RE in CNCC. $R_{ij}$ was the expression correlation of $i$-th TF and $j$-th TG across diverse ENCODE samples. $I_{kj}$ represented the interaction strength between $k$-th RE and $j$-th TG, which was learned from the PECA model on diverse cellular contexts[23]. In detail, PECA model predicted a set $D_j$ of REs to regulate the $j$-the TG. Then a regression model of $j$-th gene expression on its REs' accessibility was constructed,

$$TG_j = I_{0j} + \sum_{k \in D_j} I_{kj} RE_k \quad (2)$$

We obtained the parameter $I_{kj}$ by regression of the Eq. (2) with 148 public paired expression and accessibility data of a human (Supplementary Data 4).

The CRM associated with a TF-TG pair was defined as a set of REs bound by TF to regulate TG. We introduce $C_{ij}$ to denote a CRM for the $i$-th TF to regulate $j$-th TG and formally index the $K$ REs $RE_1, RE_2, …, RE_K$ according to their position in the genome from 5' direction. We then mathematically represented CRM $C_{ij}$ by a binary vector with length $L_{ij}$ ranging from start base of $RE_1$ to the end base of $RE_K$. For a single base $q$ in the $C_{ij}$ of $j$-th TG, we defined,

$$C_{ijq} = \begin{cases} 1 & q \in RE_1 \cup RE_2 \cup \cdots \cup RE_K \\ 0 & q \notin RE_1 \cup RE_2 \cup \cdots \cup RE_K \end{cases} \quad (3)$$

In this way, we represented the output regulatory network for a single replicate by $(S_{ij}, C_{ij})_{1 \le i \le M; 1 \le j \le N}$ with $M$ TFs and $N$ TGs, where $S_{ij}$ was regulatory strength for $i$-th TF and $j$-th TG and $C_{ij}$ was CRM linking $i$-th TF and $j$-th TG (Fig. 1).

**Consensus optimization model to integrate replicates**. Given $M$ TFs and $N$ TGs, a regulatory network was defined by the trans-regulation score $S_{1 \le i \le M; 1 \le j \le N}$ and CRM $C_{1 \le i \le M; 1 \le j \le N}$ by PECA2 procedure. For $R$ biological replicates in general, we had $R$ regulatory networks and the $r$-th was represented by $(S^r, C^r)$. In order to integrate those replicates and obtained reproducible REs and TF-TG

regulations, we proposed the following consensus optimization:

$$\min_{S,C} Q = \sum_{i,j,r} \omega_r \left\{ \left( S_{ij} - S_{ij}^r \right)^2 + \alpha \| C_{ij} - C_{ij}^r \|_2^2 \right\} - \beta \sum_{i,j} S_{ij} \| C_{ij} \|_1 - \gamma \sum_{i,j,q} C_{ijq} C_{ij(q-1)} + \mu \| S \|_1 \quad (4)$$

$$s.t.\ S_{ij} \ge 0;\ C_{ijq} \in \{0,1\} \text{ for } 1 \le i \le n, 1 \le j \le m, 1 \le q \le L_{ij};\ C_{ij0} = 0$$

Here $S, C$ were the decision variables for optimal regulatory strength and CRM. The first term in the objective function was the error term for consensus regulatory strength $S$ and consensus CRM $C$ from $R$ replicates ($S^r, C^r$). It should be minimized. $\alpha$ was to balance the scale of regulatory strength and CRM. $\omega_r$ was the weight assigned to the $r$-th biological replicate, which can be determined by replicate quality, sequencing depth, prior knowledge, or equal weight as default. The second term was maximizing consistency between regulatory strength and length of CRM. The larger the regulatory strength was, the longer the CRM was, i.e., more REs were selected in the final CRM. The third term was maximized to encourage the continuity of bases in the RE along the genome. The last term was minimized to obtain a sparse network by sparsity regularization. $\beta, \gamma, \mu$ were the parameters introduced to balance the four terms in the objective function. The constraints required positive regulatory strength $S$ and binary vector $C$ for genome position.

Model (3) was a 0-1 integer programming and known as NP-hard problem. We relaxed the integer constraints as follows,

$$S_{ij} \ge 0;\ C_{ijq} \in [0,1] \text{ for } 1 \le i \le n, 1 \le j \le m, 1 \le q \le L_{ij};\ C_{ij0} = 0 \quad (5)$$

This made Model (3) quadratic programming. And its first-order optimality conditions were:

$$\frac{\partial Q}{\partial S_{ij}} = \sum_{r=1}^{R} 2\omega_r \left( S_{ij} - S_{ij}^r \right) - \beta \| C_{ij} \|_1 + \mu = 0 \quad (6)$$

$$\frac{\partial Q}{\partial C_{ijq}} = 2\alpha \sum_{r=1}^{R} \omega_r \left( C_{ijq} - C_{ijq}^r \right) - \beta S_{ij} - \gamma \left( C_{ij(q-1)} + C_{ij(q+1)} \right) = 0 \quad (7)$$

Then we got:

$$S_{ij} = \frac{1}{2\omega} \left( 2 \sum_{r=1}^{R} \omega_r S_{ij}^r + \beta \| C_{ij} \|_1 - \mu \right) \quad (8)$$

$$C_{ijq} = \frac{1}{\omega} \sum_{r=1}^{R} \omega_r C_{ijq}^r + \frac{\beta}{2\omega\alpha} S_{ij} + \frac{\gamma}{2\omega\alpha} \left( C_{ij(q-1)} + C_{ij(q+1)} \right) \quad (9)$$

where $\omega = \sum_{r=1}^{R} \omega_r$.

We proposed the following iterative algorithm by the above optimality conditions.

**Step 1**. Initiation: assigning an initial value to $S_{ij}$ and every base $q$ in $C_{ij}$.

**Step 2**. Updating the $\hat{S}_{ij}$ and $\hat{C}_{ij}$ in last round:

 2.1 $S_{ij} = \frac{1}{2\omega} (2 \sum_{r=1}^{R} \omega_r S_{ij}^r + \beta \| \hat{C}_{ij} \|_1 - \mu)$

 2.2 $C_{ijq} = \frac{1}{\omega} \sum_{r=1}^{R} \omega_r C_{ijq}^r + \frac{\beta}{2\omega\alpha} \hat{S}_{ij} + \frac{\gamma}{2\omega\alpha} (\hat{C}_{ij(q-1)} + \hat{C}_{ij(q+1)})$

 2.3 Stop if $|S_{ij} - \hat{S}_{ij}| < \varepsilon$; else $\hat{S}_{ij} = S_{ij}, \hat{C}_{ij} = C_{ij}$

**Step 3**. Output $C_{ij}$ and $S_{ij}$.

**Extracting dense TF network from hReg-CNCC**. We supposed there were $N_0$ TFs in Module 1 and Module 2. We first extracted TF-TF regulatory matrix by taking $N_0$ TFs corresponding rows and columns, forming a $N_0 \times N_0$ TF regulatory strength matrix $S^0$. Then dense TF network was detected from this TF regulatory strength matrix by the following quadratic programming:

$$\max_{u,v} \sum_i \sum_j S_{ij}^0 u_i v_j$$

$$s.t. \sum_i u_i^2 = 1; \sum_j v_j^2 = 1; u_i \ge 0; v_j \ge 0 \quad (10)$$

The variable $u_i$ was the measure of the importance of $i$-th TF and $v_j$ for $j$-th TF. After solving this quadratic programming, the $i$-th TF was called upstream TF if $u_i \ge \mu_c$ and $v_i < v_c$; core TF if $u_i \ge \mu_c$ and $v_i \ge v_c$; downstream TF if $u_i < \mu_c$ and $v_i \ge v_c$. We took cutoffs $\mu_c = 0.1$ and $v_c = 0.05$ in our experiments.

**Subnetwork extraction from hReg-CNCC by genes, SNPs, and REs**. For gene $j$ in a gene set $J$, we checked all TF-CRM-TG triplets and retained the triplet if gene $j$ was included as a TG. Combining these triplets resulted the subnetwork of hReg-CNCC associated with a gene set. Formally, the sub-network was:

$$\bigcup_{j \in J} \left( S_{ij}, C_{ij} \right) \quad (11)$$

For SNP $p$, we checked all TF-CRM-TG triplets and retained the triplet if the SNP was located in its CRM. Then the subnetwork of hReg-CNCC associated with a set of SNPs $P$ was obtained as:

$$\bigcup_{p \in P, C_{ijp}=1} \left( S_{ij}, C_{ij} \right) \quad (12)$$

For RE $e$, we checked all TF-CRM-TG triplets and retained the triplet if its CRM

was intersected with this RE. Then, the subnetwork of hReg-CNCC associated with a set of REs $E$ was defined as follows:

$$\bigcup_{e \in E, e \cdot C_{ij} > 0} \left( S_{ij}, C_{ij} \right) \tag{13}$$

After obtaining the sub-networks, we conducted a filtering procedure to obtain smaller but more significant sub-networks. For a sub-network, we considered the TGs one by one. For a TG, we normalized its CRMs' regulatory strength to z-score and deleted the CRMs whose z-score was smaller than a threshold $\lambda$. $\lambda$ was set to be 1.5 in Fig. 3c, 0 in Figs. 4c and 6b.

**Human biased enhancer, human biasedly expressed genes, and fold change enrichment**. CNCC of human and chimpanzee were derived by iPS differentiation in vitro and human biased enhancers were fetched from (Prescott, et al.)[21]. A gene was called a human biased expressed gene if it satisfied two conditions: (1) $t$-test $P$-value $\leq 0.05$ between human and chimpanzee; (2) the average expression was higher in human than in chimpanzee.

The fold change enrichment of human biasedly expressed genes was computed by the following formula:

$$F = \frac{N_{pb}/N_p}{N_b/N} \tag{14}$$

where $N_{pb}$ was the number of predicted human biased expressed genes. $N_p$ was the number of predicted genes. $N_b$ was the number of human biased expressed genes. $N$ was the number of all genes.

A larger fold change meant higher accuracy of predicting CRM-TG regulations. The proximity-based nearest genes were detected by GREAT with the default setting. The ABC model was conducted with CNCC ATAC-seq, ChIP-seq of H3K27ac, and public averaged Hi-C data with default cutoff 0.02 as described in https://github.com/broadinstitute/ABC-Enhancer-Gene-Prediction.

**Fold change enrichment of facial shape-associated SNPs on REs**. For each SNP, we took the minimum $P$-value of 78 distances as its $P$-value. Then for thresholds 1.0, $10^{-1}$, $10^{-2}$, $10^{-3}$, $10^{-4}$, $10^{-5}$, $10^{-6}$, $10^{-7}$, $10^{-8}$, $10^{-9}$. we got SNP sets of different thresholds. For SNP set of each threshold, we defined the fold change enrichment as follows given the chromosome region set,

$$FC = \frac{P_r/L_r}{P/L} \tag{15}$$

Where $P_r$ was the number of SNPs in given chromosome region. $L_r$ was the length of the given chromosome region. $P$ was the total number of SNPs. $L$ was the genome length.

We calculated and compared $FC$ value for three kinds of chromosome region set: REs of hReg-CNCC, all union ATAC-seq peaks of CNCC, and peaks of other tissues (Supplementary Data 4).

**Naïve methods to integrate replicates**. We compared our consensus optimization-based replicate integration with two naïve methods, which were union and intersection network. Union network was defined by selecting TF-TG pair if this pair was contained by at least one of the 6 CNCC single replicate networks. The intersection network was defined by selecting the TF-TG pair if this pair was contained by all the 6 CNCC single replicate networks.

**Null model of expected overlapping number between hReg-CNCC and UCE or HAR**. We generated 10,000 random sequence sets for UCEs and HARs respectively for construction of null model. Taking UCEs for example, every random set of UCEs was generated with command "bedtools shuffle -i UCEs.bed -g hg19.sizes". The 10,000 random sets were regarded as the null model. We intersected every of 10,000 random sequence sets of UCEs with REs in hReg-CNCC and obtained the number of overlapped sequences. We found only 74 UCE random sets had more than 5 overlapped sequences with hReg-CNCC, which give the $P$-value 0.0074 of UCEs' overlapping with hReg-CNCC. Similarly, we obtained the $P$-value of the overlapping between hReg-CNCC and HARs.

**Statistics and reproducibility**. The comparison between consensus optimization and single networks in Fig. 2a was conducted by unpaired one-tailed $t$-test with $N = 6$ experiments for both single networks and consensus optimization. The heatmap and clustering were conducted with R package "pheatmap". The comparison between the dense network and random generated network with same node number was conducted by unpaired one-tailed $t$-test with $N = 19$ edges for dense network and random generated network. The overlapping between modules of hReg-CNCC and dense network, the overlapping between hReg-CNCC and hReg-CNCC-H9 was conducted with hypergeometric test with $N = 25,268$ genes. Empirical $P$-values was obtained to evaluate the overlapping between UCE (HAR) and hReg-CNCC by a randomly generated null model.

**Reporting summary**. Further information on research design is available in the Nature Research Reporting Summary linked to this article.

## Data availability

Source data underlying the main figures are presented in Supplementary Data 5. The RNA-seq and ATAC-seq data of CNCC were downloaded under GEO accession GSE70751. GWAS summary statistics of facial distance were downloaded from GWAS catalog under accession GCST009464. Human ultra-conserved elements were downloaded at https://users.soe.ucsc.edu/~jill/ultra.html. Human accelerated regions were downloaded as the supplementary data of Hubisz et al.[58]. All other data are available from the authors upon reasonable request.

## Code availability

All codes for consensus optimization and analysis were available at https://github.com/AMSSwanglab/hReg-CNCC and archived in Zenodo[73].

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

## Acknowledgements

This work was supported by Shanghai Municipal Science and Technology Major Project (No. 2017SHZDZX01), CAS "Light of West China" Program (No.xbzg-zdsys-201913), the "CAS Interdisciplinary Innovation Team" project, National Key R&D Program of China (No. 2017YFC0908400 and 2020YFA0712402), and the National Natural Science Foundation of China (NSFC) under Grants Nos. 12025107, 11871463, 61621003, and 91651507, the Strategic Priority Research Program of Chinese Academy of Sciences under Grant No. XDC01000000 and XDB38010400. Z.X. was supported by China Scholarship Council PhD Fellowship. The computations were partly done by the high-performance computers of State Key Laboratory of Scientific and Engineering Computing, Chinese Academy of Sciences. The work of W.H.W. and D.Z. was supported by NIH grants P50-HG007735 and R01HG010359. We thank reviewers for their insightful suggestions to improve the manuscript.

## Author contributions

Y.W., W.H.W., and F.L. conceived and supervised the project. Z.F. designed the analytical approach and performed numerical experiments and data analysis. Z.D. contributed to analysis of annotating facial shape-associated GWAS, human facial traits/diseases, and human evolutionary elements. Z.X. and S.W. contributed to analysis of annotating facial shape-associated GWAS. All authors wrote, revised, and contributed to the final manuscript.

## Competing interests

The authors declare no competing interests. We confirmed all the participants provided verbal informed consent to take part in the study.
