## [Peer Review File · Communications Biology]

Reviewers' comments:

Reviewers #2/6 (Remarks to the Author):

Review for "Reconstructing human regulatory network of cranial neural crest cell and annotating variants in developmental context"

In this paper, Feng et al. utilize previously published expression and chromatin accessibility data to reconstruct a gene regulatory network for cranial neural crest cells, termed hReg-CNCC. This network links transcription factor binding to regulatory elements to regulated gene expression and proposes a hierarchical architecture of transcription factors in a CNCC gene regulatory network. The authors further explore the cis regulatory sequences in their network and overlap them with GWAS SNPs, disease traits and evolutionarily interesting regions (both deeply conserved and rapidly evolving). The authors make some interesting observations in their analysis of available datasets, regarding regulation of CNCC enhancers and overlap with human variants. However, we believe the authors somewhat overstate a number of observations, and the biological relevance of some conclusions is unclear. Furthermore, some key details and comparisons for the model itself need to be provided. We outline major and minor concerns below.

Major concerns

The authors should provide more detail as to how the interaction strength score in their model is calculated, they simply state "which was learned from the PECA model on diverse cellular contexts." I could not find more details on this in the provided reference, and in any case given that it is important to the predictions, those details should be explicitly provided in this paper.

Page 1. The authors claim that in their hReg-CNCC network, transcription factors "hierarchically regulate the neural plate border, specification, and migration". The data they use to generate their network is from a single stage of in vitro CNCC differentiation. Therefore, this broad statement is unfounded, and should be updated.

Page 3. The authors perform PECA2 on data from "paired RNA-seq and ATAC-seq data to obtain *R* context-specific regulatory networks" – is it a requirement of the analysis that the RNA-seq and ATAC-seq datasets are from matched samples? If so, did the authors confirm that the samples analysed are indeed matched? If this is not a requirement, perhaps rephrase as this is confusing.

On page 4, the authors state that "Some TFs were less understood in CNCC and we labeled them as novel TFs (Figure 3B)". Many of these TFs have clear links to craniofacial development from human Mendelian disorders and from mouse genetics (e.g. ALX1/3/4, PRRX1 etc.) The authors need to reclassify these TFs.

Also page 4, using the human-biased enhancers and human-biased genes as a "gold standard" is a reasonable approach, but the authors should also compare their method to the Activity-by-contact (ABC) model (Fulco et al, Nature Genetics 2019) of predicting enhancer-gene links, as this method also outperforms the nearest gene approach. All the input data to apply the ABC method to CNCCs (ATAC-Seq, H3K27ac ChIP-seq, optionally RNA-seq) are available.

Regarding the classification of TFs into Modules 1 and 2, can the authors verify that this is not driven by something as trivial as the information content of the TFs binding site motifs? Low information content motifs would be predicted in many/most accessible regions, which would result in broad predicted regulation, whereas higher information content would result in a more specific set of predicted targets.

In many places the authors state that a TF regulates a certain gene, e.g. page 7 "TFAP2A regulated

19/28 of the causal genes; NR2F1 regulated 16/28 of the causal genes; and ALX3/4 regulated 22/28 of the causal genes". The authors don't have any direct functional evidence for these causal relationships, and so should take care to reword this through the text.

In two instances, the authors discuss that a HOX binding site was implicated as an influential motif at a regulatory element. It should be noted that the head region is for the most-part HOX-negative, and the authors should discuss which factors may bind to these sequences that may be relevant for craniofacial development, else the relevance of these binding motifs may be in question. For example, on page 7 "On the distal regulatory element, the most important one was "HOXA4_1/encode", which was involved with 18 TFs' binding" and page 8 "This regulation may be achieved by SNP's influence on the motif binding of HOXB2." Furthermore, expression data for TFs is included in the author's model – given that HOX genes are very lowly expressed (< 1 TPM) in CNCCs, this suggests a false positive prediction of the model -- a TF cannot regulate a target gene if it is not expressed. The authors may consider adding a hard expression cutoff for TFs and target genes to their predictions to eliminate these cases.

The authors state "There were two types of pleiotropy: the first was that one SNP locating in a regulatory element and regulated multiple genes. For example, rs16985457 was located in chr19:54693360-54695240 and regulated CNOT3, PRPF31, and LENG1 (Figure 4E); the second type was one SNP regulates a gene and was associated with multiple traits." It is unclear how the authors determined that these SNPs regulate these genes without functional data. Perhaps this is simply a misunderstanding and the authors should reword to clarify they mean the regulatory element regulates the target gene, and activity may be modulated by the SNP. Furthermore, in the second example the 'multiple traits' are in fact all related craniofacial measurements, and so probably does not count as a case of pleiotropy.

The statement "helped find causal SNPs, causal genes, and explained how genetic variants get involved in regulation" is an over-reach about the conclusions, and should be re-worded.

There are numerous additional concerns regarding the analysis of facial GWAS SNPs, regardless of the precise dataset used (note there is a recent bioRxiv preprint with a larger set of genome-wide significant SNPs, <https://doi.org/10.1101/2020.05.12.090555>). The overall enrichment of facial GWAS SNPs within hReg-CNCC regions is weak and seems highly dependent on the GWAS p-value threshold used -- why does Figure 4A only go up to $-\log_{10}(p)$ of 5 (axes should be labeled on this figure) instead of the standard $5e-08$ genome-wide significance cutoff? Additionally, the enrichment analysis in 4A should be repeated using a set of random SNPs not associated with facial measures, that would ideally be matched to the true set of SNPs in terms of minor allele frequency, number of additional SNPs in LD, and density of nearby genes. This matching can easily be done with tools such as SNPSnap (https://data.broadinstitute.org/mpg/snpsnap/match_snps.html).

The sentence in the Discussion "The human face is an exemplar complex morphological structure resulting from the intricate coordination of genetic, cellular, and environmental factors" bears a striking resemblance to a sentence in the above-mentioned recent bioRxiv preprint, which reads: "The human face is an exemplar complex morphological structure. It is a highly multipartite structure resulting from the intricate coordination of genetic, cellular, and environmental factors." Perhaps this was an honest mistake by the authors, but this kind of copy-pasting of sentences without citation is not acceptable.

Minor concerns

In the introduction the authors mention "Wilderman, et al. profiled multiple biochemical markers of chromatin activity as a comprehensive functional genomics data and predicted chromatin states for 4.5-8 post-conception weeks of early human craniofacial development²²" Was this data used in generating the network? This should be clarified.

The authors validate their model using GRN from model organisms "hReg-CNCC predicted 703 regulations among the 50 CNCC genes and 36 were in the known CNCC pathways.". Recently Tatjana Sauka-Spengler and colleagues expanded the neural crest GRN, and could be incorporated into the analysis (<https://doi.org/10.1016/j.devcel.2019.10.003>).

In Figure 3C, the authors should also provide the corresponding recall values for each of the TF ChIP-Seq datasets.

Axes are missing for Figure 3E.

Please provide a null model of the number of expected overlaps between hReg-CNCCs and either UCes or HARs - is the observed overlap more than this?

Please name the VISTA elements that were tested and overlap with UCes in your analysis (Figure 6B). One element was stated to be positive, but in which tissues the enhancer was active was not stated, please clarify. Also, just because a sequence was tested in VISTA does not mean that it is necessarily important (as the authors state) - many sequences tested in VISTA were also chosen because they are highly conserved, this is circular logic.

Please reword the statement "These results showed that hReg-CNCC helped find the conserved elements that were responsible for human face and illustrated their Regulations." By definition, an ultraconserved element is exactly the same between humans and other species, and so cannot be responsible for the "human face" as it is unique from other species -- in fact it is more likely that it is responsible for aspects of craniofacial development shared between species.

The importance or relevance of these statements is unclear, please provide some interpretation of this observation: "We also noticed that the TFs regulating the five annotated regulatory elements were consistent with hReg-CNCC revealed regulatory architecture. For example, TFAP2B, ALX4, and TCF4 regulated 4/5 of the annotated regulatory elements and they were in Module 1 and upstream or core TFs in dense network. TWIST1 and TFAP4 only regulated one of the five annotated regulatory elements and they were in Module 2 or downstream TFs in dense network."

"TFs in Module 1, such as TFAP2A and NR2F133, were at higher level and broadly regulated other genes." The authors should clarify what they mean by level here - there could be confusion they mean expression level, or level in the TF GRN hierarchy.

For the facial GWAS section, please provide the number of SNP-TF-TG links at a 5e-08 genome-wide significance threshold for the GWAS SNPs.

What do the authors mean in this section about 'one paired data', this is unclear. Please clarify in the text. "PECA was successfully applied to identify master regulator in stem cell differentiation²⁴ and interpret regulatory element for non-model organism²⁵. PECA2 further extended PECA to require one paired data (i.e., one sample) as input to infer the regulatory network and was applied to reveal causal regulations for time course data²⁶."

The English is difficult to understand in places and could benefit from proof-reading.

There are numerous typos, including:

- evolutionary
- optimizaiton
- intermediate

Reviewer #4 (Remarks to the Author):

Authors of Feng et al present hReg-CNCC, a network-based modeling tool that allows them "to annotate genetic variants of human facial GWAS and disease traits with associated cis- regulatory modules, transcription factors, and target genes." Linking distal regulatory regions to target genes is a major challenge for the chromatin and neural crest communities. Thus, the tools presented here could be quite useful. Additionally, hReg-CNCC has the potential to link human variation data from GWAS studies with functional data from genomics studies, enabling investigators to better understand how subtle difference in phenotype arise from modest impacts on gene regulation. This is a very interesting concept as it pertains to human evolution and sexual selection in the context on neural crest development. While there are many interesting observations in this study, which highlight the utility of hReg-CNCC, the study lacks sufficient validation of predictions. This shortcoming diminishes my enthusiasm.

Major concerns:

The main issue deals with validation - authors perform several methods attempting to validate their results, including studies of GWAS SNPs at identified REs, and measurements of TFs from bonafide NCC derived datasets (Prescott et al.). In my view, these attempts fall short, especially with consideration to the more rigorous validation performed by these research groups in prior publications (Duren 2017 and 2020). More rigorous validation methods are necessary to assess the overall utility of hReg-CNCC.

Was hReg-CNCC performed on data from ESCs and differentiated embryoid bodies as in Duren et al 2020 (Genome Research), or was new data acquired from other sources as in Duren et al 2017 (PNAS)? This is a critical issue that has not been made clear in the manuscript. If the former, how is it possible that NCC specific REs and TFs are active in ESCs and/or in embryoid bodies? Shouldn't these factors be inactive in highly heterogeneous embryoid bodies, especially at stages prior to NCC specification? If the later, it is important for the reader to understand which published datasets were used for hReg-CNCC.

The ability to recreate the hierarchy of NCC-specific TFs using their network-based modeling approach is quite impressive. The authors should further validate their results using data from a recently published study - Long et al. (Cell Stem Cell 2020), where human cells were differentiated into NCCs and then chondrocytes. If the identified REs and TFs are indeed active during NCC differentiation these factors should be active in the Long et al datasets. As an orthogonal approach, data from Long et al could be analyzed independently using PECA2 and hReg-CNCC and the results could be compared with current findings to assess similarities and differences between tissue types or datasets.

Several of the REs identified appear to be inaccessible in the portion of the genome the authors selected to depict in figure 5. This raises the question - how many REs are accessible vs. inaccessible in total for the data used in this study? Are these regions marked by H3K27ac? Are the in-accessible REs accessible in other NCC-derived ATAC-Seq datasets? Authors should investigate these questions both to validate their results, and also to assess how REs might be utilized differently in distinct NCC-derived tissues. These types of comparisons were performed in Duren et al 2017, and were quite useful for assessing the performance of PECA. It would be especially interesting to assess whether REs become active or inactive in NC-derived cancer types where master NC regulators such as Sox9 or Sox10 are known to function. Publicly available TCGA data should allow the authors to investigate this question.

Minor concerns:

Authors should confirm abbreviations are defined the first time they are used. "RE" is defined only in the methods section.

In the introduction section "Marcos et al." should be "Simoes-Costa et al."

Point-to-point responses to **Reviewers' comments:**

Reviewer #2/6 (Remarks to the Author):

In this paper, Feng et al. utilize previously published expression and chromatin accessibility data to reconstruct a gene regulatory network for cranial neural crest cells, termed hReg-CNCC. This network links transcription factor binding to regulatory elements to regulated gene expression and proposes a hierarchical architecture of transcription factors in a CNCC gene regulatory network. The authors further explore the cis regulatory sequences in their network and overlap them with GWAS SNPs, disease traits and evolutionarily interesting regions (both deeply conserved and rapidly evolving). The authors make some interesting observations in their analysis of available datasets, regarding regulation of CNCC enhancers and overlap with human variants. However, we believe the authors somewhat overstate a number of observations, and the biological relevance of some conclusions is unclear. Furthermore, some key details and comparisons for the model itself need to be provided. We outline major and minor concerns below.

Author's Response: We thank the reviewer for the precise summary. We are happy that the reviewer appreciated our major contribution in construction of human regulatory network of cranial neural crest cells (hReg-CNCC) and application to interpret GWAS SNPs, diseases, and evolutionarily important regions. In the light of the reviewer's suggestion, we presented more validations of hReg-CNCC, added complete details of our model and comparison, reworded the many statements of observation and conclusion to make the biological relevance between them clearer. We hope these revisions will address the major and minor concerns and highlight our contribution to construct high-quality human regulatory network of cranial neural crest cells and effort to interpret genetic variants, disease, and evolutionarily regulatory elements.

Major concerns:

1.) The authors should provide more detail as to how the interaction strength score in their model is calculated, they simply state "which was learned from the PECA model on diverse cellular contexts." I could not find more details on this in the provided reference, and in any case given that it is important to the predictions, those details should be explicitly provided in this paper.

Author's Response: We followed your suggestions to add more detail of how the interaction strength score was calculated in our revision. This will make the computation of interaction strength score clear. The interaction strength was based on the output of PECA model¹, which gave as set D_j of REs for the j -th TG. We conducted a regression model of j -th gene expression (TG_j) on its REs' accessibility (RE_k):

$$TG_j = I_{0j} + \sum_{k \in D_j} I_{kj} RE_k \quad (1)$$

We obtained the parameter I_{kj} , which was the interaction strength score in our manuscript, by regression of the equation (1) with 148 public paired expression and accessibility data of human. We have explicitly added these details into our "Methods" section and we thank the reviewer to improve our manuscript.

Excerpt from Manuscript: (Page 13) R_{ij} was the expression correlation of i -th TF and j -th TG across diverse ENCODE samples. I_{kj} represented the interaction strength between k -th RE and j -th TG,

which was learned from the PECA model on diverse cellular contexts¹. In detail, PECA model predicted a set D_j of REs to regulate the j -th TG. Then a regression model of j -th gene expression on its REs' accessibility was constructed,

$$TG_j = I_{0j} + \sum_{k \in D_j} I_{kj} RE_k \quad (2)$$

We obtained the parameter I_{kj} by regression of the equation (2) with 148 public paired expression and accessibility data of human (**Table S4**)

2.) Page 1. The authors claim that in their hReg-CNCC network, transcription factors “hierarchically regulate the neural plate border, specification, and migration”. The data they use to generate their network is from a single stage of in vitro CNCC differentiation. Therefore, this broad statement is unfounded, and should be updated.

Author's Response: We followed your suggestion to update the broad statement in a more specific way. We agreed with the reviewers that our hReg-CNCC were constructed with a single stage of in vitro CNCC differentiation. In our regulatory network, we found that a variety of TFs were involved with regulation of CNCC. And functions of these TFs were association with neural plate border (such as MSX1), specification (such as TWIST1), and migration (such as SOX5/9/10). The function association did not causally indicate that regulation of neural plate border, specification, and migration happened in single stage of CNCC. We have revised the sentences involved with this problem.

Excerpt from Manuscript:

(Page 1) Consensus optimization predicts high quality regulations and reveals the architecture of upstream, core, and downstream transcription factors that are associated with functions of neural plate border, specification, and migration

(Page 11) Importantly the architecture that upstream, core, and downstream TFs were associated with functions of neural plate border, specification, and migration were explored in hReg-CNCC and

3.) Page 3. The authors perform PECA2 on data from “paired RNA-seq and ATAC-seq data to obtain R context-specific regulatory networks” – is it a requirement of the analysis that the RNA-seq and ATAC-seq datasets are from matched samples? If so, did the authors confirm that the samples analysed are indeed matched? If this is not a requirement, perhaps rephrase as this is confusing.

Author's Response: We clarified the sample match issue to make this point clear. For PECA2, the paired samples were required for better quality of regulatory network. However, the samples can be matched at different level, such as biosample, replicate, or cell type level depending on the application. For example, we tried matched samples at both biosample and cell type level in mouse ENCODE data in our previous PECA studies (Duren et al. 2017). In this study, the samples were matched at biosample level. The matched samples and their GEO accession were listed in detail as below in Table R1. However, “RNAseq_human1_rep2” doesn't have a matched biosample “ATAC_human1_rep2”. Then we confirmed that there are high correlations between “RNAseq_human1_rep2” and “RNAseq_human1_rep1”. We made the adjustment about the matching condition to match “RNAseq_human1_rep2 (GSM1817213)” with “ATAC_human1_rep1 (GSM1817203)” in our study.

Pair	RNA-seq	ATAC-seq
1	RNAseq_human1_rep1(GSM1817212)	ATAC_human1_rep1(GSM1817203)
2	RNAseq_human2_rep1(GSM1817214)	ATAC_human2_rep1(GSM1817204)
3	RNAseq_human2_rep2(GSM1817215)	ATAC_human2_rep2(GSM1817205)
4	RNAseq_human3_rep1(GSM1817216)	ATAC_human3_rep1(GSM1817206)
5	RNAseq_human3_rep2(GSM1817217)	ATAC_human3_rep2(GSM1817207)
6	RNAseq_human1_rep2(GSM1817213)	ATAC_human1_rep1(GSM1817203)

Table R1. Matched samples used for hReg-CNCC's construction.

A

RNA	H1P1	H1P2	H2P1	H2P2	H3P1	H3P2
H1P1		0.97	0.96	0.96	0.96	0.97
H1P2	0.97		0.97	0.95	0.95	0.96
H2P1	0.96	0.97		0.99	0.96	0.97
H2P2	0.96	0.95	0.99		0.96	0.96
H3P1	0.96	0.95	0.96	0.96		0.98
H3P2	0.97	0.96	0.97	0.96	0.98	

B

ATAC	H1P1	H2P1	H2P2	H3P1	H3P2
H1P1		0.90	0.87	0.80	0.84
H2P1	0.90		0.94	0.74	0.79
H2P2	0.87	0.94		0.80	0.80
H3P1	0.80	0.74	0.80		0.88
H3P2	0.84	0.79	0.80	0.88	

Figure R1. (A). Correlation of 6 samples of RNA-seq. (B). Correlation of 5 samples of ATAC-seq.

As shown in Table R1, the first 5 pairs were perfectly matched at biosample level (from the same human and the same replicate) and the sixth pair was weakly matched (only from the same human). After processing all the RNA-seq and ATAC-seq, we found that there were high correlations among samples and the samples from the same human were more correlated than that of different humans (Figure R1A, B). With this observation, we assumed that if the sample of “ATAC_human1_rep2” was available, it would be quite correlated with “ATAC_human1_rep1”. So, to make the best use of all available data, we made “RNA_human1_rep2” and “ATAC_human1_rep1” the sixth paired samples.

In our revision, we added the information of sample matching into the supplement tables to avoid misunderstanding.

Excerpt from Manuscript: (Page 3) In the first step, we collected paired RNA-seq and ATAC-seq data from (Prescott, *et al.*)²¹ and applied PECA2²⁶ to R replicates ($R = 6$ in this study, **Table S1**) to obtain R context-specific regulatory networks

4.) On page 4, the authors state that “Some TFs were less understood in CNCC and we labeled them as novel TFs (Figure 3B)”. Many of these TFs have clear links to craniofacial development from human Mendelian disorders and from mouse genetics (e.g. ALX1/3/4, PRRX1 etc.) The authors need to reclassify these TFs.

Author's Response: Thanks for the reviewer's crucial comments. We have corrected this notation in our revision and labeled these TFs as “other CNCC TFs”.

Excerpt from Manuscript: (Page 5) Some TFs were not included in CNCC pathway but also important for CNCC development. So, we labeled them as “other CNCC TFs” (Figure 3B)

5.) Also page 4, using the human-biased enhancers and human-biased genes as a “gold standard” is a reasonable approach, but the authors should also compare their method to the Activity-by-contact (ABC) model (Fulco et al, Nature Genetics 2019) of predicting enhancer-gene links, as this method also outperforms the nearest gene approach. All the input data to apply the ABC method to CNCCs (ATAC-Seq, H3K27ac ChIP-seq, optionally RNA-seq) are available.

Author’s Response: We thank the reviewers and follow your valuable suggestion to compare hReg-CNCC with ABC model from (Fulco et al, Nature Genetics 2019). Our side-by-side comparison results in Figure R2 showed hReg-CNCC outperformed ABC model in predicting enhancer-gene links.

ABC (Activity-by-contact) model was based on ABC score by integrating ATAC-seq peaks, H3K27ac activity, and Hi-C contact,

$$ABC\ score_{E,G} = \frac{A_E C_{E,G}}{\sum_{All\ elements\ e\ with\ in\ 5Mb\ of\ G} A_e C_{e,G}}$$

Where A_E was the activity of enhancer and $C_{E,G}$ was the contacts between enhancer and gene. Unluckily, Hi-C data was not available for CNCC. To apply ABC model to CNCC dataset, we followed the instruction at the ABC software website (<https://github.com/broadinstitute/ABC-Enhancer-Gene-Prediction>) to leverage average Hi-C data. Totally, ABC model predicted 17,499 pairs of enhancer-gene regulations.

We first checked whether ABC model could predict human biased genes for human biased enhancers. 17 human biased enhancers were assigned target genes by ABC model and there were totally 260 genes regulated by them. Only 38 of these 260 target genes were human biased genes, which gave the fold change 1.60. This result indicated a similar performance with proximity-based method. hReg-CNCC performed better than ABC model and proximity-based method (Figure R2A). After comparison among three-method-predicted human biased genes, we found 15 genes that were uniquely detected by hReg-CNCC, which were important for CNCC (Figure R2B). For example, *ROBO3* was only expressed by human (FPKM 2.50 in human and 0.71 in chimpanzee). There was a human biased enhancer (supported by H3K27ac and ATAC-seq data) located at 65k downstream of *ROBO3* (Figure R2C). While this enhancer was closer to *HEPACAM*, it was not associated with *HEPACAM* since this gene was not expressed in human CNCC (FPKM 0.08). This example also showed the strength of hReg-CNCC to correctly detected distal enhancers for target genes.

The comparison with ABC model further confirmed that hReg-CNCC can better predict enhancer gene interaction by utilizing the rich information of gene expression from RNA-seq data. Also, hReg-CNCC didn’t require Hi-C data which is currently limited by resolution, high cost, and high demand of cell materials. We discussed the possible reason for the relatively poor performance of ABC model by inputting the averaged Hi-C data. We expected the high resolution CNCC Hi-C data may improve its accuracy and hReg-CNCC can easily incorporate the high-quality physical interaction data. We added this comparison in our revised manuscript and we thanked the reviewers’ advice to better show the advantage of hReg-CNCC.

Figure R2. (A) hReg-CNCC predicts the human biased enhancers' target genes more accurately than ABC model and proximity-based method using human biased differentially expressed genes as gold standard. (B). hReg-CNCC predicts 33% enhancer gene relationships as novel distal regulation for human biased enhancers, which cannot be found by ABC model or the proximity-based method. (C). hReg-CNCC predicts *ROBO3* as the target gene for the distal human biased enhancer (comparing human and chimpanzee's H3K27ac and ATAC-seq tracks), which is located near *HEPACAM*. The expression pattern of *ROBO3* supports the target assignment for the human biased enhancer (comparing human and chimpanzee's RNA-seq tracks).

Excerpt from Manuscript: (Page 4)We utilized the linkages between human biased enhancers²¹ and human biasedly expressed genes as gold standard positives. For these human biased enhancers, hReg-CNCC predicted 216 genes as their target genes, of which 45 genes were human biasedly expressed genes. This gave a fold change enrichment 2.31 (Methods). We compared with Activity-By-Contact (ABC) model²⁸ and proximity-based method, which assigns the nearest TSS as target gene (Methods). For ABC

model, there were 260 genes that were predicted to be regulated by human biased enhancers and 38 of them were human biasedly expressed genes, which gave the fold change 1.60. For proximity-based method, there were 1,445 nearest genes linked to these human biased enhancers and 214 genes of them were human biasedly expressed genes, which gave the fold change 1.62. (Figure 2D). These results showed hReg-CNCC was more accurate to assign correct target genes for regulatory elements. Importantly, 15 hReg-CNCC predicted human biasedly expressed genes (33%) were regulated by distal enhancers and cannot be correctly predicted by ABC model or proximity-based method (Figure 2E). For example, *ROBO3*, which confines early neural crest cells to the ventral migratory pathway in the trunk²⁹ and regulates the production of cranial neural crest cells³⁰, was predicted as the true target gene of a distal human biased enhancer, which was located near *HEPACAM* and far from *ROBO3*'s gene body (Figure 2F). This distal human biased enhancer was validated by human specific ATAC-seq and H3K27ac ChIP-seq signals and was consistent with the expression pattern of *ROBO3* (FPKM 2.50 in human and 0.71 in chimpanzee, Figure 2F). Though this human biased enhancer was nearest to *HEPACAM*, it was not associated with *HEPACAM* since *HEPACAM* was not expressed in human CNCC (FPKM 0.08)

6.) Regarding the classification of TFs into Modules 1 and 2, can the authors verify that this is not driven by something as trivial as the information content of the TFs binding site motifs? Low information content motifs would be predicted in many/most accessible regions, which would result in broad predicted regulation, whereas higher information content would result in a more specific set of predicted targets.

Author's Response: We thank the reviewers to pinpoint a possible bias for our module detection in hReg-CNCC. We calculated the information content of 32 TFs in Module 1 and 71 TFs in Module 2 but found no significant difference between their information content (Figure R3A). In fact, there existed some TFs in Module 1 that had high information content, such as *TFAP2A*, and there was also some TFs in Module 2 who had low information content, such as *LMX1B*. The main difference between Module 1 and Module 2 was the number of bound REs (Figure R3B). We also check whether information score was association number of predicted binding REs in Reg-CNCC and obtained a low correlation (PCC=0.08, Figure R3B). These results showed that the classification of TFs into Module 1 and Module 2 was not driven by trivial information content.

Figure R3. (A) Boxplot of information content (IC score) of TFs in Module 1 and Module 2. (B). Scatter plot of the IC score and number of hReg-CNCC-predicted bound REs of 103 TF (32 Module 1 TFs and 71 Module 2 TFs) show no correlation between IC score and binding REs.

Excerpt from Manuscript: (Page 5)the clustering was not driven by motif's trivial information content (Figure S2A, B)

7.) In many places the authors state that a TF regulates a certain gene, e.g. page 7 “TFAP2A regulated 19/28 of the causal genes; NR2F1 regulated 16/28 of the causal genes; and ALX3/4 regulated 22/28 of the causal genes”. The authors don’t have any direct functional evidence for these causal relationships, and so should take care to reword this through the text.

Author’s Response: We thank the reviewers to point out the lack of functional evidence for the “causal” relationship in our statements. The TF-TG regulations were inferred from hReg-CNCC. We have reworded the sentences involved with the misleading “causal” relationship in our revision.

Excerpt from Manuscript:

(Page 7)The genetic variants in the CRMs of hReg-CNCC, including their functional regulatory elements, **target genes**, and bound TFs, should be useful in the annotation of SNPs identified by GWAS of human facial variation traits.....

(Page 8)we found that even though different traits at different region of face had different SNPs and target genes, they shared a group of upstream TFs. For example, TFAP2A regulated 19/28 of the **target genes**; NR2F1 regulated 16/28 of the target genes; and ALX3/4 regulated 22/28 of the **target genes**.....Together these evidences suggested ALX1 as the **candidate facial shape associated gene** in our annotated regulatory network.....

(Page 9)hReg-CNCC can improve the enrichment of facial shape-associated SNPs in CNCC and used the TFs, REs, and TGs to help explain how genetic variants get involved in regulation, such as ALX1.....This fact motivated us depict human facial traits’ the known genetic variants by its target genes and regulation.....

8.) In two instances, the authors discuss that a HOX binding site was implicated as an influential motif at a regulatory element. It should be noted that the head region is for the most-part HOX-negative, and the authors should discuss which factors may bind to these sequences that may be relevant for craniofacial development, else the relevance of these binding motifs may be in question. For example, on page 7 “On the distal regulatory element, the most important one was “HOXA4_1/encode”, which was involved with 18 TFs’ binding” and page 8 “This regulation may be achieved by SNP’s influence on the motif binding of HOXB2.” Furthermore, expression data for TFs is included in the author’s model – given that HOX genes are very lowly expressed (< 1 TPM) in CNCCs, this suggests a false positive prediction of the model -- a TF cannot regulate a target gene if it is not expressed. The authors may consider adding a hard expression cutoff for TFs and target genes to their predictions to eliminate these cases.

Author’s Response: We agree with the reviewer that revealing true upstream binding TF is difficult and we thank the reviewers for the suggestion about considering the expression of motif associated TFs. The HOX TFs were indeed non- or low-expressed in CNCC. So, we reexamined the motifs these two SNPs might exert influence on. On the distal RE of ALX1, if we change the allele at SNP rs11609649 to its effective allele, there were a gain of “PH0082.1_Irx2/Jaspar” and a loss of “FOXM1_1/encode”, corresponding to IRX3 and FOXM1. IRX3 and FOXM1 were highly expressed in CNCC (RPKM of IRX3 68.12, FOXM1 49.03) and important regulators of CNCC^{2,3}. On the promoter, the change to SNP rs12810608’s effective allele will cause a gain of “PB0186.1_Tcf3_2/Jaspar”, which was associated with TCF cluster, such as TCF3, TCF7L1, TCF12, and TCF4 (Figure R4). We revised this paragraph in our manuscript.

Figure R4. Two SNPs influencing the regulation of ALX1 and their located REs, influenced TFs.

Excerpt from Manuscript: (Page 8)We scanned motif on these two REs with effective allele and reference allele of the SNPs respectively and found that the binding affinity of many motifs were changed (Figure 4D). There was a gain of motif “PB0186.1_Tcf3_2/Jaspar” in promoter and its associated TCF clusters were top regulators of ALX1 (Figure S4B). On the distal regulatory element, when allele at rs11609649 was change to the effective allele, there were a gain of motif “PH0082.1_Irx2/Jaspar” and a loss of motif FOXM1_1/encode. These two motifs corresponded to IRX3 and FOXM1 respectively, which were highly expressed in CNCC (FPKM of IRX3 68.12, FOXM1 49.03) and important regulators of CNCC^{45,46}

9.) The authors state “There were two types of pleiotropy: the first was that one SNP locating in a regulatory element and regulated multiple genes. For example, rs16985457 was located in chr19:54693360-54695240 and regulated CNOT3, PRPF31, and LENG1 (Figure 4E); the second type was one SNP regulates a gene and was associated with multiple traits.” It is unclear how the authors determined that these SNPs regulate these genes without functional data. Perhaps this is simply a misunderstanding and the authors should reword to clarify they mean the regulatory element regulates the target gene, and activity may be modulated by the SNP. Furthermore, in the second example the ‘multiple traits’ are in fact all related craniofacial measurements, and so probably does not count as a case of pleiotropy.

Author’s Response: We agree with the reviewer that claiming regulations without functional data should be careful. In our revision, we clarified the relationship between SNPs and target gene are inferred from paired omics data by hReg-CNCC. They mean the regulatory element regulates the target gene, and activity may be modulated by the SNP, i.e., the SNPs were located in the regulatory elements and exerted influence on the activity of regulatory elements. And these regulatory elements were predicted to regulate target genes by hReg-CNCC. We also corrected the misuse of “pleiotropy” in our revision. The multiple facial distances were quite related and couldn’t be

classified to be “pleiotropy”. Instead, we termed this pattern as “multi-trait effect” in our revised manuscript.

Excerpt from Manuscript:

(Page 8)We revealed several interesting patterns to potentially illustrate SNPs’ multi-trait effect and cooperation. There were two patterns of **multi-trait effect: the first was that one SNP was located in a regulatory element and this regulatory element regulated multiple genes**. For example, rs16985457 was located in chr19:54693360-54695240 and chr19:54693360-54695240 was predicted to regulate CNOT3, PRPF31, and LENG1 (Figure 4E, left); **the second type was one SNP, which was associated with multiple traits, was located in a regulatory element and this regulatory element only regulates a gene**. For example, rs12810608 was located in chr12:85673460-85674718, which regulates ALX1, and this SNP was associated with three face distances: “EnR-Prn”, “EnL-Prn”, and “EnR-All” (Figure 4E, middle). In addition, **multiple SNPs cooperated in one regulatory element and worked together to influence the activity of regulatory element**. For instance, rs11719548 and rs11711710 were simultaneously located in chr3:12872024-127872868 and chr3:12872024-127872868 regulated RUVBL1 (Figure 4E, right).....

(Page 9)hReg-CNCC can also illustrate the possible mechanism of SNPs’ **multi-trait effect** and cooperation.....

10.) The statement “helped find causal SNPs, causal genes, and explained how genetic variants get involved in regulation” is an over-reach about the conclusions, and should be re-worded.

Author’s Response: We removed the over-reaching statement in our revision.

Excerpt from Manuscript: (Page 9)hReg-CNCC can improve the enrichment of facial shape-associated SNPs in CNCC and used the TFs, REs, and TGs to help explain how genetic variants get involved in regulation.....

11.) There are numerous additional concerns regarding the analysis of facial GWAS SNPs, regardless of the precise dataset used (note there is a recent bioRxiv preprint with a larger set of genome-wide significant SNPs, <https://doi.org/10.1101/2020.05.12.090555>). The overall enrichment of facial GWAS SNPs within hReg-CNCC regions is weak and seems highly dependent on the GWAS p-value threshold used -- why does Figure 4A only go up to $-\log_{10}(p)$ of 5 (axes should be labeled on this figure) instead of the standard $5e-08$ genome-wide significance cutoff? Additionally, the enrichment analysis in 4A should be repeated using a set of random SNPs not associated with facial measures, that would ideally be matched to the true set of SNPs in terms of minor allele frequency, number of additional SNPs in LD, and density of nearby genes. This matching can easily be done with tools such as SNPSnap (https://data.broadinstitute.org/mpg/snpsnap/match_snps.html).

Author’s Response: We thank the reviewers’ comments and pointing out SNPSnap to help us to perform additional control experiments.

We agreed that the overall enrichment of facial GWAS SNPs in hReg-CNCC was weak and dependent on GWAS p-value threshold. The cutoff in Figure 4A only went up to 10^{-5} because the number of SNPs was not large enough to obtain a reliable fold change score. As described in the “Methods” section, we used FC score to evaluate the enrichment of SNPs in given region set, which was defined as follow:

$$FC = \frac{P_r/L_r}{P/L}$$

Where P_r was the number of SNPs in given region set. L_r was the length of the given region set. P was the total number of SNPs. L was the genome length. We calculated this FC score for SNPs set filtered by threshold 1.0, 10^{-1} , 10^{-2} , 10^{-3} , 10^{-4} , 10^{-5} , 10^{-6} , 10^{-7} , 10^{-8} , 10^{-9} . We hypothesized that if one GWAS was enriched in a region set, the FC score would increase along with the above thresholds and this strategy was used in many publications⁴. We applied this method to our study to make comparison with hReg-CNCC, CNCC peaks, and other tissues' peaks. We found that while the increasing trend can be observed, the FC score of all three region sets decreased when threshold $P\text{-value} < 10^{-6}$ (Figure R5A). We checked the number of SNPs of each threshold and found that when we set threshold to be 10^{-6} , the number of SNPs was only 1,762 (Figure R5A), which was not enough to calculate a genuine FC score. So, we focused on the FC score curve from 1.0 to 10^{-5} . We found that even though the enrichment level was a little weak, the FC score curve of hReg-CNCC REs and CNCC peaks had the increasing trend while the FC score curve of other tissues didn't increase along with threshold (Figure R5A). This observation indicated that Facial GWAS SNPs were more enriched in tissue of CNCC than other tissue. We also found that the slope (0.063) of hReg-CNCC RE set was larger than that of CNCC peaks (0.018) (Figure R5A). This result showed that facial GWAS SNPs were more enriched in hReg-CNCC RE set. We used this enrichment analysis to demonstrate that hReg-CNCC can improve the quality of regulatory elements by integrating accessibility and expression data.

Following the reviewers' suggestion, we generated a random SNP set for 10^{-5} threshold filtered SNP set with SNPsnap. We repeated the analysis in Figure 4A to calculate the FC score of SNPsnap generated SNPs set in hReg-CNCC RE set, CNCC peak set, and other 76 tissues' peak set. For facial GWAS SNPs, hReg-CNCC ranked the first among all 78 region sets and CNCC ranked the 15-th, showing that facial SNPs were more enriched in hReg-CNCC. And for random SNPs generated by SNPsnap, neither hReg-CNCC nor CNCC ranked top among 78 region sets, where hReg-CNCC ranked 5-th and CNCC ranked 61-th. This showed that the highest FC score of facial SNPs in hReg-CNCC was not randomly generated.

We thank the reviewers for the recommendation of recently published facial GWAS⁵. With cohort of 8,246 individuals, Julie D. White and his colleagues found 17,612 SNPs that were significantly associated with segmentation of facial shape. These new discovered genetic variants were promising for more understanding of human facial variation. So, it would be interesting to apply hReg-CNCC to interpret significant SNPs found by these recent GWAS studies in the future.

Excerpt from Manuscript: (Page 12)It's not surprising that hReg-CNCC can only interpret limited number of SNPs and the overall enrichment is quite modest. The craniofacial tissues represent progressively later time points and the intermediate cell types during development and their regulatory networks should be reconstructed in future.....

Figure R5. (A). Line chart: FC score of each threshold filtered SNPs in hReg-CNCC RE set, CNCC peak set, and other tissues' peak sets; bar chart: the number of each threshold filtered SNPs. (B). Up: the ranked FC score of facial GWAS SNPs in hReg-CNCC RE set, CNCC peak set, and other 76 tissues' peak sets. Down: the ranked FC score of SNPsnap generated SNPs in hReg-CNCC RE set, CNCC peak set, and other 76 tissues' peak sets.

12.) The sentence in the Discussion “The human face is an exemplar complex morphological structure resulting from the intricate coordination of genetic, cellular, and environmental factors” bears a striking resemblance to a sentence in the above-mentioned recent bioRxiv preprint, which reads: “The human face is an exemplar complex morphological structure. It is a highly multipartite structure resulting from the intricate coordination of genetic, cellular, and environmental factors.” Perhaps this was an honest mistake by the authors, but this kind of copy-pasting of sentences without citation is not acceptable.

Author’s Response: We are sorry to miss the citation about this sentence. We have added this citation of the formal Nature Genetics publication for this bioRxiv preprint in the revised manuscript. This is a truly insightful and inspiring work.

Excerpt from Manuscript: (Page 11)The human face is an exemplar complex morphological structure resulting from the intricate coordination of genetic, cellular, and environmental factors⁶³.....

63. White, J.D. et al. Insights into the genetic architecture of the human face. Nature Genetics (2020).

Minor Comments:

1.) In the introduction the authors mention “Wilderman, et al. profiled multiple biochemical markers of chromatin activity as a comprehensive functional genomics data and predicted chromatin states for 4.5-8 post-conception weeks of early human craniofacial development²²” Was this data used in generating the network? This should be clarified.

Author’s Response: We clarified in our revision that that we didn’t use this dataset and we have added this information in the overview of our method. This dataset was not used in construction of hReg-CNCC and all the data we used were from (Prescott, et al.)⁶.

Excerpt from Manuscript: (Page 3)In the first step, we collected paired RNA-seq and ATAC-seq data from (Prescott, et al.)²¹ and applied PECA2²⁶ to R replicates (R=6 in this study, Table S1) to obtain R context-specific regulatory networks.....

2.) The authors validate their model using GRN from model organisms “hReg-CNCC predicted 703 regulations among the 50 CNCC genes and 36 were in the known CNCC pathways.”. Recently Tatjana Sauka-Spengler and colleagues expanded the neural crest GRN, and could be incorporated into the analysis (<https://doi.org/10.1016/j.devcel.2019.10.003>).

Author’s Response: We thank the reviewers’ suggestion to add recently published neural crest GRN as another validation of hReg-CNCC. This further demonstrate that hReg-CNCC can predict gene regulation with better accuracy.

Tatjana Sauka-Spengler and colleagues profiled NC-specific epigenomic (ATAC-seq, H3K27ac ChIP-seq) and transcriptomic data in chick. By comparing with non-NC tissue, the NC-specific enhancer candidates were detected. Based on this NC-specific enhancer cluster, the authors constructed GRN for chick neural crest. This GRN of chick could be a reference for human neural crest. Following the suggestion of the reviewers, we conducted supplement validation with this chick GRN as gold standard and obtained the same conclusion as before. First, we compared hReg-CNCC with six original network and we found a significant outperformance for metric of precision, recall and F1 score (Figure R6A). Then we made comparison with intersection and union methods. hReg-CNCC performed best for precision and ranked second for recall. The F1 score also revealed that hReg-CNCC was best among three methods (Figure R6B). These results again showed that hReg-CNCC can predict gene regulation with best accuracy and satisfying coverage. We thank the reviewers and these results were added into our revised manuscript.

Figure R6. Comparison with chick-GRN as gold standard. (A). Consensus optimization achieves significantly higher precision, recall, and F1 measure than single networks. (B). Consensus optimization outperforms the naive union and intersection methods in precision, recall, and F1 measure.

Excerpt from Manuscript: (Page 4)Then we collected another GRN as gold standard for parallel validation, which was built with multi-omics data in chick²⁷. We reached the same conclusion as with CNCC pathway: hReg-CNCC was significantly better than single networks for precision, recall, and F1 score (Figure S1E). Compared with overlapping and union method, hReg-CNCC obtained the best precision,

with a trade-off of recall. And hReg-CNCC performed best for F1 score (Figure S1F), which again showed hReg-CNCC was the best among three methods.....

3.) In Figure 3C, the authors should also provide the corresponding recall values for each of the TF ChIP-Seq datasets.

Author’s Response: We thank the reviewers about the suggestion to add recall value in Figure 2C. For both recall and precision, hReg-CNCC outperformed the 6 single regulatory networks for TFAP2A and NR2F2 ChIP-seq validation (Figure R7).

Figure R7. Precision and recall of NR2F1 and TFAP2A ChIP-seq validation.

4.) Axes are missing for Figure 3E.

Author’s Response: We thank the reviewer’s comment. We have added the axes for Figure 3E.

5.) Please provide a null model of the number of expected overlaps between hReg-CNCCs and either UCES or HARs - is the observed overlap more than this?

Author’s Response: We thank the reviewers about the suggestion to provide a null model of overlap with UCES and HARs. To do this, we generated 10,000 random sequence sets for UCES and HARs respectively. Every random set of UCES was generated with command “bedtools shuffle -i UCES.bed -g hg19.sizes” and every random set of HARs was generated with command “bedtools shuffle -i HARs.bed -g hg19.sizes”. The random sets were regarded as the null model. We intersected every of 10,000 random sequence sets of UCES with REs in hReg-CNCC and obtained the number of overlapped sequences. We found only 74 random sets of UCES had more than 5 overlapped sequences with hReg-CNCC, which gives P-value 0.0074 of UCES’ overlapping with hReg-CNCC. This indicated that the overlapping between hReg-CNCC and UCES was significant. Similarly, we obtained the significance (P-value=0.1001) of the overlapping between hReg-CNCC and HARs. We have added this information in our manuscript.

Excerpt from Manuscript: (Page 10)We found five regulatory elements in hReg-CNCC were overlapped with human UCES (P-value<0.0074, Method, Figure 6B)In total, 13 regulatory elements in hReg-CNCC were found to be associated with HAR (P-value<0.1001, Figure S4A)

6.) Please name the VISTA elements that were tested and overlap with UCES in your analysis (Figure 6B). One element was stated to be positive, but in which tissues the enhancer was active was not stated, please clarify. Also, just because a sequence was tested in VISTA does not mean that it is necessarily important

(as the authors state) – many sequences tested in VISTA were also chosen because they are highly conserved, this is circular logic.

Author's Response: We added the additional information about the name and active tissue for VISTA enhancers into Figure 6B. We agree with the reviewers that “a sequence was tested in VISTA does not mean that it is necessarily important” and we re-worded the statements here to avoid the circular logic.

Excerpt from Manuscript: (Page 10)Among them, three elements were also candidate enhancers in VISTA database⁵³. And one of these three VISTA enhancers “chr8:77690693-77691421” was positive for transgenic mouse assay (4/4 were limb positive and 1/4 was neural tube positive), showing its possible role in neural crest.....

7.) Please reword the statement “These results showed that hReg-CNCC helped find the conserved elements that were responsible for human face and illustrated their Regulations.” By definition, an ultraconserved element is exactly the same between humans and other species, and so cannot be responsible for the “human face” as it is unique from other species -- in fact it is more likely that it is responsible for aspects of craniofacial development shared between species.

Author's Response: We thank the reviewers to point out the problem about conclusion of UCE analysis. The UCEs were almost the same among different species, which made them the important elements. Here we overlapping UCEs with regulatory elements in CNCC and found 5 UCEs were linked to CNCC. The conservation among species and activity in CNCC prioritized these five UCEs to be important regulatory elements for neural crest and facial development among species. We reworded this sentence in the revised manuscript.

Excerpt from Manuscript: (Page 10)These results showed that hReg-CNCC helped find the conserved elements that were responsible for **facial development** and illustrated their regulations.....

8.) The importance or relevance of these statements is unclear, please provide some interpretation of this observation: “We also noticed that the TFs regulating the five annotated regulatory elements were consistent with hReg-CNCC revealed regulatory architecture. For example, TFAP2B, ALX4, and TCF4 regulated 4/5 of the annotated regulatory elements and they were in Module 1 and upstream or core TFs in dense network. TWIST1 and TFAP4 only regulated one of the five annotated regulatory elements and they were in Module 2 or downstream TFs in dense network.”

Author's Response: We are sorry about the unclear statements here. The point we wanted to make here was the accordance between architecture of UCEs associated sub-network and whole hReg-CNCC network. In hReg-CNCC, we found two modules of TFs: Module 1 TFs were in higher level of network and broadly bound on regulatory elements and regulated other genes; Module 2 TFs were responsible for more specific regulations and regulated a small group of genes. The TFs in the UCE associated regulatory network also revealed this architecture. TFAP2B, ALX4, and TCF4 were Module 1 TFs in hReg-CNCC. In the UCE associated network, they bound on 4 of the 5 UCEs, which agreed with their broad regulation property. TWIST1 and TFAP4 were Module 2 TFs in hReg-CNCC. In the UCE regulatory network, they only bound on one of the five UCEs and regulated fewer genes. This was also consistent with their specific regulation property. We have rewritten these statements to make our point clearer.

Excerpt from Manuscript: (Page 10)In the UCE associated subnetwork, we noticed that there were two types of TFs that were consistent with hReg-CNCC revealed 2-Module regulatory architecture. For example, TFAP2B, ALX4, and TCF4 represented the first type of TFs and they regulated 4 of the 5 annotated UCEs. Their property of broad regulation in UCE network agreed with the fact that they were in Module 1 and upstream or core TFs in dense network. On the other hand, TWIST1 and TFAP4 represented the second type TFs and they only regulated one of the five annotated UCEs, showing their feature of specific regulation. This was in accordance with the fact that they were in Module 2 or downstream TFs in dense network.....

9.) “TFs in Module 1, such as TFAP2A and NR2F133, were at higher level and broadly regulated other genes.” The authors should clarify what they mean by level here - there could be confusion they mean expression level, or level in the TF GRN hierarchy.

Author’s Response: We are sorry we did not make this meaning of “level” clear. The “level” indicated the level of regulatory network hierarchy. We rewrote this sentence in the revised manuscript.

Excerpt from Manuscript: (Page 7)TFs in Module 1, such as TFAP2A and NR2F1³³, were at higher level of regulatory network and broadly regulated other genes.....

10.) For the facial GWAS section, please provide the number of SNP-TF-TG links at a 5e-08 genome-wide significance threshold for the GWAS SNPs.

Author’s Response: We thank the reviewers’ suggestion. When we set the threshold of SNPs to be 5e-8, we found no SNPs were overlapped with REs in hReg-CNCC. Instead we used 1e-5 as the relaxed threshold.

11.) What do the authors mean in this section about ‘one paired data’, this is unclear. Please clarify in the text. “PECA was successfully applied to identify master regulator in stem cell differentiation²⁴ and interpret regulatory element for non-model organism²⁵. PECA2 further extended PECA to require one paired data (i.e., one sample) as input to infer the regulatory network and was applied to reveal causal regulations for time course data²⁶.”

Author’s Response: We are sorry that we did not make the concept of “paired data” clear. “One paired data” means one sample with match RNA-seq and ATAC-seq data. We revised it in our manuscript.

Excerpt from Manuscript: (Page 2)PECA2 further extended PECA by removing the requirement of paired data from a diverse panel of cell types, so that inference of context specific regulatory network is possible from paired expression and chromatin accessibility data on just one sample.....

12.) The English is difficult to understand in places and could benefit from proof-reading.

There are numerous typos, including:

- evolutionary
- optimizaiton
- intermediate

Author's Response: We thank the reviewers pinpoint the typos and the problem of English. We corrected these typos and rewrote parts of our manuscript that were difficult to understand.

Reviewer #4 (Remarks to the Author):

Authors of Feng et al present hReg-CNCC, a network-based modeling tool that allows them "to annotate genetic variants of human facial GWAS and disease traits with associated cis- regulatory modules, transcription factors, and target genes." Linking distal regulatory regions to target genes is a major challenge for the chromatin and neural crest communities. Thus, the tools presented here could be quite useful. Additionally, hReg-CNCC has the potential to link human variation data from GWAS studies with functional data from genomics studies, enabling investigators to better understand how subtle difference in phenotype arise from modest impacts on gene regulation. This is a very interesting concept as it pertains to human evolution and sexual selection in the context on neural crest development. While there are many interesting observations in this study, which highlight the utility of hReg-CNCC, the study lacks sufficient validation of predictions. This shortcoming diminishes my enthusiasm.

Author's Response: We are happy the reviewer appreciated our major contribution to construct a regulatory network of human CNCC, which gives the TF-CRM-TG regulations and can hypothesize how genetic variants get involved in regulatory network and affect phenotypes. We also thank the reviewer's comments about the problem of validation. In light of the reviewers' suggestion, we include more datasets for validation and comparison to obtain more solid conclusion.

1. The main issue deals with validation - authors perform several methods attempting to validate their results, including studies of GWAS SNPs at identified REs, and measurements of TFs from bonafide NCC derived datasets (Prescott et al.). In my view, these attempts fall short, especially with consideration to the more rigorous validation performed by these research groups in prior publications (Duren 2017 and 2020). More rigorous validation methods are necessary to assess the overall utility of hReg-CNCC.

Author's Response: We thank the reviewer to point out the limitation of validation of hReg-CNCC in our manuscript. Following the suggestion of the reviewer, we conducted more validation and comparison to show the accuracy and outperformance of hReg-CNCC. First, we used a most recently published GRN of CNCC of chick as reference to validate the TF-TG regulation in hReg-CNCC and comparison with other methods. Again, we showed the higher accuracy of hReg-CNCC. Then we used the H3K27ac ChIP-seq data to validate the REs in hReg-CNCC and we obtained 74.42% precision. Some regulation of RE to TG can be validated by independent Capture-C dataset. For example, the REs that were predicted to regulate SOX9 were contacted by loops to promoter of SOX9. Then we performed comprehensive comparison of accuracy of RE-TG regulation in hReg-CNCC with ABC model and hReg-CNCC showed the better performance. For the two-module architecture of hReg-CNCC, we added comparison of TFs' information content to show that the classification of these two modules was biologically meaningful. Finally, we constructed regulatory using the same consensus optimization with recent published independent CNCC dataset. We found that the TF-RE-TG regulations and two-module architecture were quite reproducible. We hope these supplement comparisons can solve the reviewer's concern of validation.

2. Was hReg-CNCC performed on data from ESCs and differentiated embryoid bodies as in Duren et al 2020 (Genome Research), or was new data acquired from other sources as in Duren et al 2017 (PNAS)? This is a critical issue that has not been made clear in the manuscript. If the former, how is it possible that

NCC specific REs and TFs are active in ESCs and/or in embryoid bodies? Shouldn't these factors be inactive in highly heterogeneous embryoid bodies, especially at stages prior to NCC specification? If the later, it is important for the reader to understand which published datasets were used for hReg-CNCC.

Author's Response: We were sorry that we did not make the dataset we used clear. We used the recently-published CNCC dataset for the construction of hReg-CNCC. In detail, paired RNA-seq and ATAC-seq data of CNCC were fetched from (Prescott, *et al.*)⁶ and used for hReg-CNCC network reconstruction.

Excerpt from Manuscript: (Page 3)In the first step, we collected paired RNA-seq and ATAC-seq data from (Prescott, *et al.*)²¹ and applied PECA2²⁶ to R replicates (R=6 in this study, Table S1) to obtain R context-specific regulatory networks.....

3. The ability to recreate the hierarchy of NCC-specific TFs using their network-based modeling approach is quite impressive. The authors should further validate their results using data from a recently published study - Long *et al.* (Cell Stem Cell 2020), where human cells were differentiated into NCCs and then chondrocytes. If the identified REs and TFs are indeed active during NCC differentiation these factors should be active in the Long *et al.* datasets. As an orthogonal approach, data from Long *et al.* could be analyzed independently using PECA2 and hReg-CNCC and the results could be compared with current findings to assess similarities and differences between tissue types or datasets.

Author's Response: We enormously thank the reviewer's suggestion to utilize the recent CNCC dataset to validate our results. Long *et al.* (Cell Stem Cell 2020) indeed provides a valuable dataset and we formally cited this work in our revision. Our computation in this dataset leads to hReg-CNCC-H9 and it persuasively validated our results as follows and greatly improved our manuscript.

Following the suggestion of the reviewer, we constructed another regulatory network of CNCC, named hReg-CNCC-H9, using 4 RNA-seq replicates and 4 ATAC-seq replicates at passage 4 stage during hESC's differentiation to neural crest. We chose this stage because it was the closest stage to hReg-CNCC. In total, there were 399 TFs, 9,146 TGs and 3,1145 REs in hReg-CNCC-H9. We first check the similarity of these two independent regulatory networks. We found a significant overlap of TF set, TG set, RE set, and TF-TG regulations between these two networks (Figure R8A). This indicated that the identified TFs, REs, and TGs were also active in (Long. *et al.*) dataset and they were indeed important regulators during neural crest differentiation. Second, we checked if the architecture of hReg-CNCC was reproducible in hReg-CNCC-H9 network. Again, we can also find two modules in the new CNCC regulatory network (Figure R8B). Module 1 was marked the its 18 TFs, which broadly regulated most of the TGs. For Module 2, there were 76 TFs and they were responsible for much specific regulation. Furthermore, we observed a significant similarity between two modules in hReg-CNCC-H9 network and modules in hReg-CNCC (Figure R8C). These results showed that the two-module architecture of hReg-CNCC was reproducible in other datasets and indeed biologically meaningful for neural crest. Finally, we used the dense network strategy to obtain the hierarchy of TFs in hReg-CNCC-H9 (Figure R8D). There was some difference between dense TF network of hReg-CNCC-H9 and hReg-CNCC. For example, the upstream TFs were different. We also found the hierarchy of hReg-CNCC-H9 had common characteristics with hReg-CNCC. For example, the upstream and core TFs, which were higher level of regulatory network, were largely shared, including TFAP2A/B, ALX1/3/4, NR2F1, PRRX2, and MYCN. And the downstream TFs of hReg-CNCC-H9 and hReg-CNCC were also overlapped, such as TWIST1, SIX1, TCF7L1, LMX1B, and SOX4. These showed that the hierarchy of NCC-specific TFs was also reproducible in other datasets. In summary, our hReg-CNCC identified active TFs, REs, and TGs for CNCC, found two-module architecture and hierarchy of NCC-specific TFs for CNCC regulatory network, which were

validated by an independent and valuable dataset in a recently published study - Long et al. (Cell Stem Cell 2020).

On the other hand, we noticed that there was Capture-C data for SOX9 in (Long, *et al.*) dataset to provide physical chromatin interactions. We used this data to validate the regulation of SOX9 in hReg-CNCC. There were two REs that were predicted to regulate SOX9: one was on the promoter of SOX9 and the other was in the 45k downstream of SOX9. The distal REs was contacted by a loop with promoter of SOX9, demonstrating the accuracy of our prediction of RE-TG regulation (Figure R8E). We made comparison with ABC model, which obtain RE-TG regulation by ABC score. There were 6 REs that were predicted by ABC model to regulate SOX9 and only one of them can be validated by loops of Capture-C. ABC model also failed to identify the regulation of SOX9's promoter. This supplement evidence showed that hReg-CNCC can reveal more accurate enhancer-target gene regulation.

We thank the reviewer again and we added these validations into our revised manuscript.

Excerpt from Manuscript:

(Page 6)To evaluate the reproducibility of hReg-CNCC and its hierarchical architecture, we built another regulatory network (hReg-CNCC-H9) with an independent CNCC dataset³⁶. hReg-CNCC-H9 was based on the paired RNA-seq and ATAC-seq data of human H9-ESC differentiated CNCC dataset³⁶ and was reconstructed with the same consensus optimization model of hReg-CNCC. First, we found significant overlapping of TFs, TGs, REs, and TF-TG regulations between hReg-CNCC and hReg-CNCC-H9 (Figure S3A), revealing these genes and REs were indeed active in CNCC context. Second, we found that there were also two modules in hReg-CNCC-H9 (Figure S3B): TFs in the first module broadly regulated most of the TGs and were significantly shared with Module 1 TF in hReg-CNCC (Figure S3C, P-value \leq 5.11e-22); the regulations of TFs in the second module were much more specific and was significantly overlapped with Module 2 TFs in hReg-CNCC (Figure S3C, P-value \leq 1.65e-40). This indicated that the two-module architecture of hReg-CNCC was reproducible. Third, we obtained the dense TF network of hReg-CNCC-H9 as we did for hReg-CNCC. We observed consistence of hierarchy of TFs between hReg-CNCC and hReg-CNCC-H9 (Figure S3D). For example, the upstream and core TFs, which were higher level of regulatory network, were largely shared, including TFAP2A/B, ALX1/3/4, NR2F1, PRRX2, and MYCN. And the downstream TFs of hReg-CNCC-H9 and hReg-CNCC were also overlapped, such as TWIST1, SIX1, TCF7L1, LMX1B, and SOX4. These results showed that the hReg-CNCC and its hierarchical architecture was well-validated and revealed the biological property of CNCC.....

(Page 4)there were some CRM-TG regulations in hReg-CNCC that can be validated by Capture-C assay. For example, two REs were predicted by hReg-CNCC to regulate SOX9. One RE was located on SOX9's promoter and the other RE was located at the 45k downstream of SOX9. It was noted that the distal RE and SOX9 were linked by a loop of Capture-C data (Figure 2G). As comparison, ABC model predicted 6 REs to regulate SOX9 and only one of them can be validated by loops of Capture-C (Figure 2G). This again show the outperformance of hReg-CNCC to predict CRM-TG regulation.....

Figure R8. (A). Overlap of TFs, TGs, REs, and TF-TG regulations between hReg-CNCC-H9 and hReg-CNCC. (B) Heatmap of hReg-CNCC-H9 reveals two-module architecture. (C) The TFs of two modules are significantly shared by hReg-CNCC-H9 and hReg-CNCC. (D) Dense TF network of hReg-CNCC-H9. (E) Integrative plot of regulation of SOX9. The upper track was Capture-C signal anchored by SOX9 promoter. The small black boxes in the middle are regulatory elements predicted to regulate SOX9. Figures are in next page.

3. Several of the REs identified appear to be inaccessible in the portion of the genome the authors selected to depict in figure 5. This raises the question - how many REs are accessible vs. inaccessible in total for the data used in this study? Are these regions marked by H3K27ac? Are the in-accessible REs accessible in other NCC-derived ATAC-Seq datasets? Authors should investigate these questions both to validate their results, and also to assess how REs might be utilized differently in distinct NCC-derived tissues. These types of comparisons were performed in Duren et al 2017, and were quite useful for assessing the performance of PECA. It would be especially interesting to assess whether REs become active or inactive in NC-derived cancer types where master NC regulators such as Sox9 or Sox10 are known to function. Publicly available TCGA data should allow the authors to investigate this question.

Author's Response: We thank the reviewer for the suggestions on the activity of REs. We agreed that some REs in Figure 5D were less accessible than some other peaks in this portion of genome, such as peak at promoter of *TBL2*. To check the accessibility of the 5 REs that regulate *BAZ1B*, we counted the ATAC-seq reads on the REs and calculated the “openness” score¹ to evaluate their accessibility, which was defined as below:

$$O = \frac{(X + \delta)/L}{(Y + \delta)/L_0}$$

Where X was the read count in this region with length L . Y was read count in the background region with length L_0 . δ was the pseudocount to avoid zero in denominator. A region with openness ≥ 2 can be viewed as an accessible region. We found that even though some REs were less accessible, they were all above 2.0 openness (Figure R9A), indicating they were accessible regions in genome. Then we accessed all REs in hReg-CNCC and we found they were all with openness ≥ 2 and median openness of about 10.0 (Figure R9B). This showed that all the REs in hReg-CNCC were accessible in CNCC context. Second, we fetched the H3K27ac peaks from 10 samples in (Prescott. *et al.*) dataset and merged them into a union set of 109,671 H3K27ac peaks. There were 15,686 REs in hReg-CNCC and 11,673 of them were overlapped with H3K27ac peaks, which is statistically significant (empirical P-value $< 1e-4$) and gave the precision of 0.74 and recall of 0.11 (Figure R9C). This showed that most of REs in hReg-CNCC were marked by H3K27ac signal, indicating their potential role of active enhancers or promoter. Since all the REs in H3K27ac were accessible, we accessed the role of ATAC-seq peaks that were not include in hReg-CNCC. There were totally 32,256 that were identified as ATAC-seq peaks but not inferred as a regulatory element in hReg-CNCC. We collected ATAC-seq data from 76 human tissues in ENCODE (Table R2) and overlapped them with these 32,256 ATAC-seq peaks (Figure R9D). We found “Thyroid gland”, “CD4 primary cells”, “CD8 primary cells”, and “Fetal thymus” were top ranked by the overlapped number with the 32,256 ATAC-seq peaks that were not include in hReg-CNCC. It was noted that “Thyroid gland” and “Thymus” were part of the neck gland, which were the derivatives of neural crest.

We agreed that it would be quite interesting to access the activity of REs in hReg-CNCC in NC-derived cancer. In light of the reviewer’s suggestion, we collected ATAC-seq peaks for SKCM (Skin Cutaneous Melanoma) of TCGA at <https://gdc.cancer.gov/about-data/publications/ATACseq-AWG>. To make comparison with normal tissue, we collected tissue of “lower leg skin” in ENCODE under accession ENCSR864IGD. We overlapped the 15,686 REs in hReg-CNCC with ATAC-seq peaks of “lower leg skin” and “SKCM” respectively. We found that 12,297 of the hReg-CNCC’s REs (78.39%) were also accessible in the normal “lower leg skin”, which indicated that the neural crest derivatives shared some of the epigenomic landscape of neural crest. However, only 1,366 REs (8.71%) in hReg-CNCC were accessible in “SKCM” (Figure R9E). And the REs that regulated important NC TFs, such as SOX9/10 as the reviewer mentioned, were all inactive in SKCM (Figure R9F). This different overlapping ratios with hReg-CNCC between normal skin and skin cancer were consistent with the huge difference between normal skin and skin cancer (Figure R3E). These observations showed that hReg-CNCC held the promise to study the pathology of cancer.

Figure R9. (A). Read counts and openness score of five REs regulating BAZ1B. (B). Distribution of openness score of 15,686 REs in hReg-CNCC. (C). Overlapping of REs in hReg-CNCC and H3K27ac peaks. (D). Overlapping of 32,256 CNCC peaks not included in hReg-CNCC and ATAC-seq peaks of 76 human tissues. (E). Overlapping of REs in hReg-CNCC, ATAC-seq peaks of “lower leg skin”, and “SKCM”. (F). The REs regulating SOX9 and SOX10 were inactive in SKCM.

Minor concerns:

1. Authors should confirm abbreviations are defined the first time they are used. "RE" is defined only in the methods section.

Author's Response: We are sorry about the missing definition of “RE” in the “Results” section. We added this definition in the revised manuscript.

2. In the introduction section "Marcos et al." should be "Simoes-Costa et al."

Author's Response: We were sorry about the typos about the citation and we corrected it in our revision.

Reference

1. Duren, Z.N., Chen, X., Jiang, R., Wang, Y. & Wong, W.H. Modeling gene regulation from paired expression and chromatin accessibility data. *Proceedings of the National Academy of Sciences of the United States of America* **114**, E4914-E4923 (2017).
2. Cain, C.J. *et al.* Loss of Iroquois homeobox transcription factors 3 and 5 in osteoblasts disrupts cranial mineralization. *Bone reports* **5**, 86-95 (2016).
3. Wang, Z.B. *et al.* FoxM1 in Tumorigenicity of the Neuroblastoma Cells and Renewal of the Neural Progenitors. *Cancer Research* **71**, 4292-4302 (2011).
4. Meuleman, W. *et al.* Index and biological spectrum of human DNase I hypersensitive sites. *Nature* **584**, 244-251 (2020).
5. White, J.D. *et al.* Insights into the genetic architecture of the human face. *Nature Genetics* (2020).
6. Prescott, S.L. *et al.* Enhancer divergence and cis-regulatory evolution in the human and chimp neural crest. *Cell* **163**, 68-83 (2015).

REVIEWERS' COMMENTS:

Reviewer #1 (Remarks to the Author):

The manuscript Feng et al. has been greatly improved during revision. I am quite satisfied with the additional analysis to validate prior results. I do have a few minor concerns which would improve the manuscript further.

1) Given the strong overlap observed from the venn diagrams within figure S3, I feel readers would find this result valuable and thus, these panels should be included as part of the main figure 3 rather than the supplement.

2) It was encouraging to see results from cancer data analysis provided in the rebuttal letter (Figure R9). These results provide strong biological and clinical support for the utility of the tools presented. I feel readers of this manuscript would value this information and the clinical context. Rather than excluding these analyses from the manuscript entirely, these data should be included as an additional supplemental figure.

3) In the discussion section, the authors should provide biological reasoning for why overlaps from venn diagrams in figure S3 are not absolute. Clearly perfect overlap is not expected, but it would be nice if the authors discussed this.

Reviewers #2-3 (Remarks to the Author):

Overall, we believe the manuscript is improved with additional analyses and clarification in the text. However, we have a few remaining comments below.

From major comments

3. The datasets appear not to be paired based on the same experiment, therefore wouldn't it be most accurate to say the data are paired based on the cell type not biosample level?

5. We were pleased to see the authors have used the ABC model to predict regulatory relationships. In the ABC model, several different threshold cutoffs can be chosen. For example, in Fulco et al a cutoff of 0.02 is used, but later studies use 0.015 when assessing GWAS SNP enrichment (perhaps the more relevant application here). Can the authors state which threshold they chose? And if stringent, try a few other thresholds?

6. In figure S2B, the authors should explain the meaning of the y axis (Number of binding REs (*1000)), as this metric seems to nicely correlate with their module 1 and 2 regions.

8. The authors make an overstatement here about the role of IRX3 and FOXM1 in CNCC biology. "These two motifs corresponded to IRX3 and FOXM1 respectively, which were highly expressed in CNCC (FPKM of IRX3 68.12, FOXM1 49.03) and important regulators of CNCC45,46 The papers the authors cite do not support this conclusion, and instead indicate that these factors play important roles in craniofacial development or in CNCC derivatives.

Could the authors also clarify if rs11609649 is also GWAS SNP?

11. The authors have still not updated Figure 4A to show the full range of GWAS p-value cutoffs. It is misleading to only show the fold-enrichment for GWAS SNPs up to $-\log_{10}(p)$ of 5. The authors say that beyond this, there are too few SNPs to calculate a "genuine FC score," but what does a genuine

FC score actually mean in this context? There is no objective criterion used to define this. If the authors are so concerned about the low number of SNPs in this analysis, then they should use the larger set of SNPs from White et al, 2021, as we suggested previously. Alternatively, they should show the full range of p-value cutoffs (i.e. replace Figure 4A with Figure R5), and explain in the text that after seeing these results, they then focused on the $-\log_{10}(p) = 5$ cutoff. Furthermore, the authors have not added the results for SNPsnap-matched random sets of SNPs to the manuscript, Figure R5 b) is only in the rebuttal document. Given that the random SNP set still shows a FC for hReg-CNCC regions above 1 (but still less than the true SNP set), this is also important to show in the manuscript as it further shows that the overall enrichment for facial GWAS SNPs is somewhat weak.

From minor comments

5. The overlap of hReg-CNCCs does not appear significant with HARs "Similarly, we obtained the significance (P-value=0.1001)". Are the authors claiming that the hReg-CNCCs are enriched at HARs?

6. The three hReg-CNCCs regions which overlap an enhancer tested in VISTA are all negative for facial reporter signal (Hs264 and Hs476 are completely negative in E11.5 embryos). This provides no evidence for craniofacial regulatory activity of these regions, and so we would recommend that the data is removed, or a statement to this effect is made - that the loci are not active in the developing facial structures at E11.5 of mouse embryonic development. The one element that was active as an enhancer is not reproducibly active in the face, and so the statement "showing its possible role in neural crest....." is not true.

10. Again, we would still ask here that the authors acknowledge in the manuscript that they have reduced the threshold, and that no SNPs overlapped with hReg-CNCC REs at $5e-8$. It is misleading to the reader if this is not stated.

Other comments

Figure numbering for Figure 4 appears to be incorrect.

Point-to-point responses to Reviewers' comments:

Reviewer #1:

1.) Given the strong overlap observed from the venn diagrams within figure S3, I feel readers would find this result valuable and thus, these panels should be included as part of the main figure 3 rather than the supplement.

Author's Response: We followed the suggestion to move the Venn diagram in Figure S3 into the Figure 3 in main text. This kind suggestion helps to show the reproducibility of hReg-CNCC and improves our manuscript.

2.) It was encouraging to see results from cancer data analysis provided in the rebuttal letter (Figure R9). These results provide strong biological and clinical support for the utility of the tools presented. I feel readers of this manuscript would value this information and the clinical context. Rather than excluding these analyses from the manuscript entirely, these data should be included as an additional supplemental figure.

Author's Response: We thank the reviewer's comments. Following the advice of the reviewer, we included the cancer data analysis into the Supplement Figure 6 and discussed these observations in our main manuscript.

Excerpt from Manuscript: (Page 13)Another possible application is about the cancer of CNCC derivatives, such as skin. We found that most of the REs of hReg-CNCC were also accessible in "lower leg skin", but inaccessible in Skin Cutaneous Melanoma (Supplementary Figure 6a). Many REs of CNCC regulators, such as SOX9/10, were also inactive in Skin Cutaneous Melanoma (Supplementary Figure 6b). These observation indicates the potential role of hReg-CNCC to study cancer of CNCC derivatives.....

3.) In the discussion section, the authors should provide biological reasoning for why overlaps from venn diagrams in figure S3 are not absolute. Clearly perfect overlap is not expected, but it would be nice if the authors discussed this.

Author's Response: We followed the suggestion of the reviewer to discuss the overlapping between hReg-CNCC and hReg-CNCC-H9 in the revised manuscript. The difference between hReg-CNCC and hReg-CNCC-H9 may result from the different biological material used: iPSC for hReg-CNCC and hESC for hReg-CNCC-H9.

Excerpt from Manuscript: (Page 7) It was noted that hReg-CNCC and hReg-CNCC-H9 were significantly but not fully overlapped, which may result from the different biological material they used (iPSC for hReg-CNCC, hESC for hReg-CNCC-H9).....

Reviewer #2/3:

From major comments

3. The datasets appear not to be paired based on the same experiment, therefore wouldn't it be most accurate to say the data are paired based on the cell type not biosample level?

Author's Response: We thank the reviewer to point out the misleading point and we followed the reviewers' suggestion to clarify that our paired data were at cell type level in the main manuscript.

Excerpt from Manuscript: (Page 3)we collected paired RNA-seq and ATAC-seq data from (Prescott, et al.)²¹ and applied PECA2²⁶ to R replicates (R=6, samples were matched at cell type level, Supplementary Data 1) to obtain R context-specific regulatory networks.....

2. We were pleased to see the authors have used the ABC model to predict regulatory relationships. In the ABC model, several different threshold cutoffs can be chosen. For example, in Fulco et al a cutoff of 0.02 is used, but later studies use 0.015 when assessing GWAS SNP enrichment (perhaps the more relevant application here). Can the authors state which threshold they chose? And if stringent, try a few other thresholds?

Author's Response: The threshold was set to be 0.02 by default in the original ABC model publication. We think this threshold was not too stringent since 17,499 enhancer-target pairs were predicted, which was considerable to hReg-CNCC's RE-TG prediction 15,686. We added this threshold information into our manuscript.

Excerpt from Manuscript: (Page 16)The ABC model was conducted with CNCC ATAC-seq, H3K27ac ChIP-seq, and public averaged Hi-C data with **default cutoff 0.02** as described in.....

6. In figure S2B, the authors should explain the meaning of the y axis (Number of binding REs (*1000)), as this metric seems to nicely correlate with their module 1 and 2 regions.

Author's Response: We are sorry for the misleading label of y axis in Figure S2B. The y axis represented the number of binding REs of TFs. For example, TFAP2A bound on 11,515 REs and LMX1B only bound on 1,393 REs. Number of binding REs showed the regulatory range of TF and was correlated with classification of Module 1 and Module 2. We relabeled the y axis in Supplementary Figure 2b to be "number of binding REs of TFs".

8. The authors make an overstatement here about the role of IRX3 and FOXM1 in CNCC biology. "These two motifs corresponded to IRX3 and FOXM1 respectively, which were highly expressed in CNCC (FPKM of IRX3 68.12, FOXM1 49.03) and important regulators of CNCC45,46

The papers the authors cite do not support this conclusion, and instead indicate that these factors play important roles in craniofacial development or in CNCC derivatives.

Could the authors also clarify if rs11609649 is also GWAS SNP?

Author's Response: We are sorry for the overstatement of the function of IRX3 and FOXM1. We have reworded this sentences. Based on the 1×10^{-5} threshold, rs11609649 was a GWAS SNP, which P-value 1.55e-06. We added this information into our revised manuscript.

Excerpt from Manuscript:

(Page 6)SNP rs12810608 (**P-value 3.30e-07**) was located in ALX1's promoter and SNP rs11609649 (**P-value 1.55e-06**) was located in a distal regulatory region.....

11. The authors have still not updated Figure 4A to show the full range of GWAS p-value cutoffs. It is misleading to only show the fold-enrichment for GWAS SNPs up to $-\log_{10}(p)$ of 5. The authors say that beyond this, there are too few SNPs to calculate a "genuine FC score," but what does a genuine FC score actually mean in this context? There is no objective criterion used to define this. If the authors are so concerned about the low number of SNPs in this analysis, then they should use the larger set of SNPs from White et al, 2021, as we suggested previously. Alternatively, they should show the full range of p-value cutoffs (i.e. replace Figure 4A with Figure R5), and explain in the text that after seeing these results, they then focused on the $-\log_{10}(p) = 5$ cutoff. Furthermore, the authors have not added the results for SNPsnap-matched random sets of SNPs to the manuscript, Figure R5 b) is only in the rebuttal document. Given that the random SNP set still shows a FC for hReg-CNCC regions above 1 (but still less than the true SNP set), this is also important to show in the manuscript as it further shows that the overall enrichment for facial GWAS SNPs is somewhat weak.

Author's Response: We followed the reviewers' suggestion to include full range of GWAS P-value cutoffs in Figure 4a. In addition, we discussed the decreasing pattern after 1×10^{-6} and then focused on the SNPs with P-value $\leq 1 \times 10^{-5}$. We also included the SNPsnap background into Supplement Figure 3 to show that true SNP set was more enriched in hReg-CNCC than random SNP set.

(Page 7)First, we observed FC score of all three region sets decreased when threshold P-value $\leq 1 \times 10^{-6}$ and this may result from the insufficient number of SNPs. There were only 1,762 SNP with threshold P-value $< 1 \times 10^{-6}$. This motivated us to focus the enrichment analysis on SNPs with P-value $\leq 1 \times 10^{-5}$And the Facial SNPs' enrichment in hReg-NCCC were higher than random SNP set generated by SNPsnap³⁸ (Supplementary Figure 3).....

From minor comments:

5. The overlap of hReg-CNCCs does not appear significant with HARs "Similarly, we obtained the significance (P-value=0.1001)". Are the authors claiming that the hReg-CNCCs are enriched at HARs?

Author's Response: We were sorry for the misleading expression here and we restated it in the manuscript.

(Page 17)Similarly, we obtained the P-value of the overlapping between hReg-CNCC and HARs.....

6. The three hReg-CNCCs regions which overlap an enhancer tested in VISTA are all negative for facial reporter signal (Hs264 and Hs476 are completely negative in E11.5 embryos). This provides no evidence for craniofacial regulatory activity of these regions, and so we would recommend that the data is removed,

or a statement to this effect is made - that the loci are not active in the developing facial structures at E11.5 of mouse embryonic development. The one element that was active as an enhancer is not reproducibly active in the face, and so the statement “showing its possible role in neural crest.....” is not true.

Author’s Response: We followed the reviewers’ advice to make statement that the loci were not active the developing facial structures at E11.5 of mouse embryonic development.

(Page 10)And one of these three VISTA enhancers “chr8:77690693-77691421” was positive for transgenic mouse assay (4/4 were limb positive and 1/4 was neural tube positive), but showing no activity in the developing facial structures at E11.5 of mouse embryo.....

10. Again, we would still ask here that the authors acknowledge in the manuscript that they have reduced the threshold, and that no SNPs overlapped with hReg-CNCC REs at $5e-8$. It is misleading to the reader if this is not stated.

Author’s Response: We clarified that no SNPs with P-value $\leq 5 \times 10^{-8}$ were overlapped with hReg-CNCC.

(Page 7)we next scanned every facial shape-associated SNPs with P-value $\leq 1 \times 10^{-5}$ in TF-CRM-TG triplet of hReg-CNCC (no SNPs with P-value $\leq 5 \times 10^{-8}$ were overlapped with hReg-CNCC).....

Other comments:

Figure numbering for Figure 4 appears to be incorrect.

Author’s Response: We thank the reviewer to point out the error of figure number and we have corrected them in the revised manuscript.